# EQUIVARIANT MASKED POSITION PREDICTION FOR EFFICIENT MOLECULAR REPRESENTATION

**Junyi An[1]\*, Chao Qu[2]\*, Yunfei Shi[1], Xinhao Liu[3], Qianwei Tang[4], Fenglei Cao[1]†, Yuan Qi[5]†**
[1]Shanghai Academy of Artificial Intelligence for Science
[2]INFLY TECH (Shanghai) Co., Ltd.
[3]School of Computer Science, Fudan University
[4]State Key Laboratory for Novel Software Technology, Nanjing University
[5]Artificial Intelligence Innovation and Incubation (AI[3]) Institute, Fudan University

## ABSTRACT

Graph neural networks (GNNs) have shown considerable promise in computational chemistry. However, the limited availability of molecular data raises concerns regarding GNNs' ability to effectively capture the fundamental principles of physics and chemistry, which constrains their generalization capabilities. To address this challenge, we introduce a novel self-supervised approach termed Equivariant Masked Position Prediction (EMPP), grounded in intramolecular potential and force theory. Unlike conventional attribute masking techniques, EMPP formulates a nuanced position prediction task that is more well-defined and enhances the learning of quantum mechanical features. EMPP also bypasses the approximation of the Gaussian mixture distribution commonly used in denoising methods, allowing for more accurate acquisition of physical properties. Experimental results indicate that EMPP significantly enhances performance of advanced molecular architectures, surpassing state-of-the-art self-supervised approaches.

## 1 INTRODUCTION

Graph neural networks (GNNs) have found widespread application in computational chemistry. However, unlike other fields such as natural language processing (NLP), the limited availability of molecular data hampers the development of GNNs in this domain. For example, one of the largest molecular dataset, OC20 (Chanussot et al., 2021), contains only 1.38 million samples, and collecting more molecular data with ab initio calculations is both challenging and expensive. To address this limitation, molecular self-supervised learning has gained increasing attention. This approach enables molecular GNNs to learn more general physical and chemical knowledge, enhancing performance in various computational chemistry tasks, such as drug discovery (Hasselgren & Oprea, 2024) and catalyst design (Chanussot et al., 2021).

Current self-supervised methods for molecular learning contain two mainstream categories: masking and denoising. Masking methods (Hu et al., 2020; Hou et al., 2022; Inae et al., 2023) adapt the concept of masked token prediction from natural language processing (NLP) to graph learning, where graph information, such as node attribute, is masked instead of token. However, there are two major limitations: underdetermined reconstruction and lack of deep quantum mechanical (QM) insight. As illustrated in Figure 1(a), (i) reconstructing attribute of the masked iodine atom can yield multiple possible solutions, and as the number of masked atoms increases, the number of solutions will increase rapidly, making it difficult for training data to cover all possibilities; (ii) the attribute of the masked carbon atom can be inferred from the 2D geometric principles of the benzene ring, causing the model to overlook essential atomic interactions needed for learning QM properties (Messiah, 2014). In contrast, denoising methods (Zaidi et al., 2023; Feng et al., 2023) are physics-informed and facilitate self-supervised learning of QM information in equilibrium structures. Their core idea involves adding noise to atomic positions and predicting them (see Figure 1(b)). In this process,

---

\*Equal contribution: Junyi An and Chao Qu. Correspondence to Junyi An: anjunyi@sais.com.cn
†Corresponding authors: Fenglei Cao and Yuan Qi
[0]Our code is released in https://github.com/ajy112/EMPP

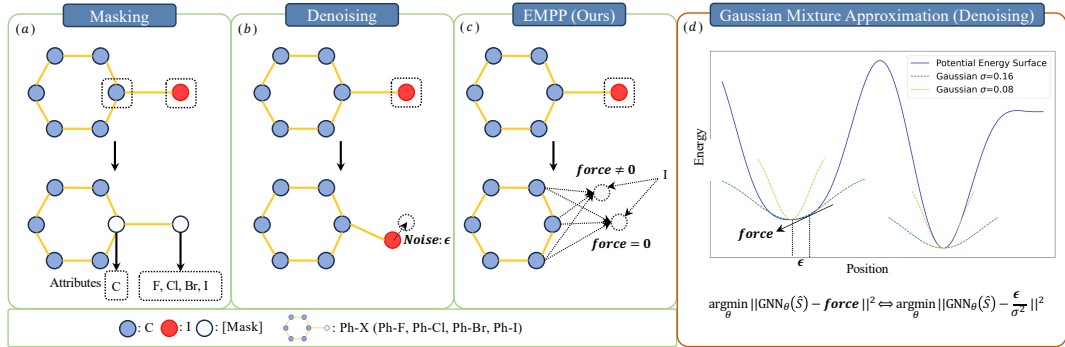

Figure 1: (a, b, c) Comparison of three molecular self-supervised methods using real halobenzenes (Ph-X) as an example: (a) Masking atomic attributes, such as atomic number, and reconstructing them; (b) Adding noise to atomic positions and predicting the noise; (c) Completely masking positions and inferring them based on neighboring structures. (d) Principle of the denoising methods: they utilize Gaussian mixture distributions to approximate the local minima of the PES, allowing the noise terms to estimate the forces, i.e. derivatives of the PES. The two local minima in the figure correspond to two equilibrium atoms, each requiring a different standard deviation $\sigma$ for approximation. As the shape of the PES is unknown, determining $\sigma$ requires an empirical approach.

the local minima of the potential energy surface (PES) are approximated using Gaussian mixture distributions (Zaidi et al., 2023), making noise prediction equivalent to learning the forces, i.e., the derivatives of the PES. However, the actual PES presents diverse and unknown local minima shapes (see Figure 1(d)), making it difficult to accurately parameterize the Gaussian mixture distribution. Our ablation studies in Section 5.4 further illustrate that model performance is sensitive to the standard deviation $\sigma$ of the Gaussian mixture. We propose addressing these limitations through a position prediction process, which allows the model to effectively learn critical QM features.

In this paper, we introduce *Equivariant Masked Position Prediction* (EMPP), a novel training method designed to enhance molecular representations in GNNs. In EMPP, we randomly mask an atom's 3D position while retaining its other attributes, such as atomic number, ensuring that position prediction remains a well-posed problem compared to attribute masking methods. Furthermore, the masked atom's position is determined by quantum mechanics within the neighboring structure, which our method is designed to learn. It is important to note that EMPP is fundamentally different from denoising methods. As illustrated in Figure 1(c), EMPP completely removes the node of the masked atom and uses the embeddings of unmasked atoms to predict its position. Consequently, EMPP can bypass the approximation of Gaussian mixture distributions, resulting in a more deterministic position prediction process, as detailed in Section 3.2. Additionally, since EMPP significantly alters the original molecular graph by removing nodes, it can generate a vast number of diverse data during training. Given a dataset of $M$ molecules with an average of $N$ atoms each, EMPP can produce $O(MN)$ well-posed examples.

EMPP is a versatile training paradigm. It enables self-supervised training of models on unlabeled equilibrium data to capture general knowledge, while also serving as an auxiliary task to connect known quantum properties with atomic positions. Experiments demonstrate that EMPP enhances the generalization of advanced equivariant GNNs (Thölke & Fabritiis, 2022; Liao & Smidt, 2023) across various molecular tasks, regardless of whether extra data is used for pre-training. Additionally, EMPP achieves state-of-the-art performance compared to previous masking and denoising methods. Our key contributions can be summarized as follows: *we propose a molecular learning paradigm that utilizes position prediction to address the challenges posed by previous masking and denoising approaches, paving a novel path for molecular learning.*

## 2 PRELIMINARIES

In this section, we briefly review the mathematical background, which includes molecular property prediction, equivariance, spherical harmonics, and so on. More detailed introductions are deferred to Appendix A. We list the notations frequently used in the following. We denote the unit sphere as $S^2$, where the spherical coordinates $(\theta, \phi)$ are the polar angle and the azimuth angle, respectively. The

symbol $\mathbb{R}$ represents the set of real numbers, while $\mathbf{R}$ represents the rotation matrix for 3D vectors. We use $SO(3)$ to denote the special orthogonal group, i.e. the 3D rotation group.

## 2.1 MOLECULAR PROPERTY PREDICTION

Molecular property prediction aims to construct a projection from the molecular 3D structure to the molecular properties. In the following, we use the $z \in \{1, ..., 118\}$ to denote the atomic number, or $\mathbf{z}$ to denote richer atomic attributes, including atomic numbers, chemical environments, and other features. We use the term $\mathbf{p}$ to denote atomic 3D position. In a $N$-nodes molecule, properties can be divided into global properties $y \in \mathbb{R}$ and node-wise properties $\mathbf{y} \in \mathbb{R}^N$. In detail, given a 3D molecular $S = \{(\mathbf{z}_i, \mathbf{p}_i)|i \in \{1, \ldots, N\}\}$, the properties can be predicted by a GNN:

$$y = \mathrm{PRED}(\mathbf{f}), \text{ and } \mathbf{f} = \mathrm{GNN}(S), \tag{1}$$

where $\mathbf{f}$ represents the node embeddings in the final GNN layer, and $\mathrm{PRED}(\cdot)$ is the prediction head.

## 2.2 EQUIVARIANCE

Given any transformation parameter $g \in G$, a function $\varphi : \mathcal{X} \to \mathcal{Y}$ is called equivariant to $g$ if it satisfies:

$$T'(g)[\varphi(x)] = \varphi(T(g)[x]), \tag{2}$$

where $T'(g) : \mathcal{Y} \to \mathcal{Y}$ and $T(g) : \mathcal{X} \to \mathcal{X}$ denote the corresponding transformations over $\mathcal{Y}$ and $\mathcal{X}$, respectively. Invariance is a special case of equivariance where $T'(g)$ is an identity transformation. In this paper, we mainly focus on the $SO(3)$ equivariance and invariance, since it is closely related to the interactions between atoms in molecule [1]. In other words. The backbone and prediction head should adhere to equation 2.

## 2.3 SPHERICAL HARMONICS AND STEERABLE VECTOR

**Spherical harmonics**, a class of functions defined over the sphere $S^2$, form an orthonormal basis and have some special algebraic properties widely used in equivariant models (Kondor et al., 2018; Cohen et al., 2018). In this paper, we use the real-valued spherical harmonics denoted as $\{Y_m^l : S^2 \to \mathbb{R}\}$, where $l$ and $m$ denote degree and order, respectively. It is known that any square-integrable function defined over $S^2$ can be expressed in a spherical harmonic basis via

$$f(\theta, \phi) = \sum_{l=0}^{\infty} \sum_{m=-l}^{l} f_m^l Y_m^l(\theta, \phi), \tag{3}$$

where $f_l^m$ is the Fourier coefficient. For any vector $\vec{r}$ with orientation $(\theta, \phi)$, we define $\mathbf{Y}^l(\vec{r}/\|\vec{r}\|) = \mathbf{Y}^l(\theta, \psi) = [Y_{-l}^l(\theta, \psi); Y_{-l+1}^l(\theta, \psi); ...; Y_l^l(\theta, \psi)]^T$, a vector with $2l+1$ elements. Furthermore, we define the spherical harmonics representation for any direction:

$$\mathrm{sh}^l(\frac{\vec{r}}{\|\vec{r}\|}) = [\mathbf{Y}^0(\frac{\vec{r}}{\|\vec{r}\|}); \mathbf{Y}^1(\frac{\vec{r}}{\|\vec{r}\|}); ...; \mathbf{Y}^l(\frac{\vec{r}}{\|\vec{r}\|})]. \tag{4}$$

It forms a $(l+1)^2$ vector. Equivariant models (Zitnick et al., 2022; Liao & Smidt, 2023; An et al., 2024) typically set a maximum degree $L_{max}$ and construct node embeddings with $C$ channels, resulting in an embedding size of $(L_{max} + 1)^2 \times C$. The full mathematical form of spherical harmonics can be found in Appendix A.1.

A commonly used property of the spherical harmonics is that for any $\mathbf{R} \in SO(3)$, we have $\mathbf{Y}^l(\mathbf{R}\vec{r}) = \mathbf{D}^l(\mathbf{R})\mathbf{Y}^l(\vec{r})$, where $\mathbf{D}^l(\mathbf{R})$ is a $(2l+1) \times (2l+1)$ matrix known as a Wigner-D matrix with degree $l$. Therefore, $\mathbf{R}$ and $\mathbf{D}^l(\mathbf{R})$ corresponds to $T(g)$ and $T'(g)$, respectively in equation 2. Following the convention in (Chami et al., 2019; Brandstetter et al., 2022), we say $\mathbf{Y}^l(\vec{r})$ is steerable by Wigner-D matrix of the same degree $l$. The $2l+1$-dimensional vector space on which a Wigner-D matrix of degree $l$ act is termed a type-$l$ steerable vector space, denoted by the superscript $(l)$ in this paper.

**Equivariant operations**. To ensure the equivariance of the entire model, each operation must maintain equivariance. The e3nn library (Geiger et al., 2022) offers common equivariant operations, including SO(3) linear transformations, SO(3) normalizations, gate activations, and Clebsch–Gordan (CG) tensor products. Passaro & Zitnick (2023) further extends certain nonlinear equivariant operations to the sphere. In our work, we leverage existing equivariant models as backbones and employ aforementioned equivariant operations to construct our position prediction block.

---

[1]Invariance of translation is trivially satisfied by taking the relative positions as inputs.

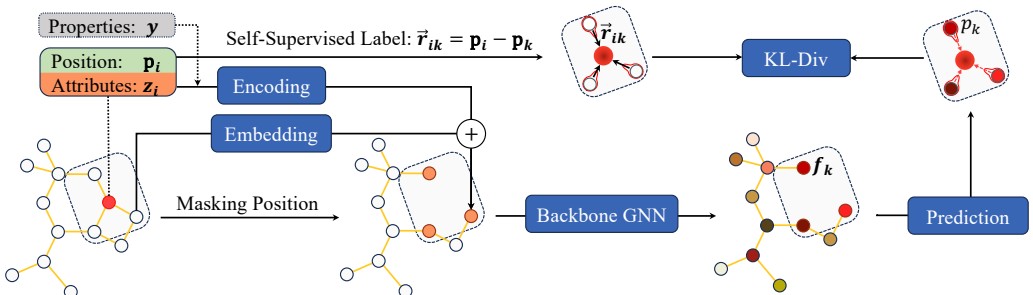

Figure 2: The overall framework of EMPP. The masked position can be recounstructed by the GNNs output features of the neighboring nodes, with the position determined by the predicted directions and radius from those nodes.

## 3 METHODOLOGY

In this section, we first revisit the existing molecular self-supervised methods and outline their primary limitations. Subsequently, we present our approach, *Equivariant Masked Position Prediction* (EMPP), and elaborate on its implementation details.

### 3.1 REVISITING THE VANILLA MASK METHOD IN MOLECULE LEARNING

Similar to NLP, molecular self-supervised methods aim to learn the underlying chemical and physical mechanisms in molecular systems, such as valence bond theory (Shaik & Hiberty, 2007) and force fields (Ponder & Case, 2003). To this end, AttrMask (Hu et al., 2020) pioneers a method that randomly masks atoms and predicts their attributes. More formally, we assume the $i, j, ...$-th atoms are masked and the modified molecule is denoted as $\hat{S} = \{(\mathbf{z}_1, \mathbf{p}_1), ..., (\mathbf{M}, \mathbf{p}_i), ..., (\mathbf{M}, \mathbf{p}_j), ..., (\mathbf{z}_N, \mathbf{p}_N)\}$, where $\mathbf{M}$ denotes the mask vector (like the [MASK] token in BERT (Devlin et al., 2019)). The attributes of the masked atoms are predicted by a GNN as follows:

$$\hat{\mathbf{z}}_{i,j,...} = \text{PRED}(\mathbf{f}_{i,j,...}), \quad \mathbf{f}_1, ..., \mathbf{f}_N = \text{GNN}(\hat{S}), \tag{5}$$

where $\mathbf{f}_{i,j,...}$ represents the GNN output features of the masked atoms, and $\text{PRED}(\cdot)$ typically refers to a neural network. The objective is to minimize the discrepancy between the predicted attributes $\hat{\mathbf{z}}_{i,j,...}$ and the actual attributes $\mathbf{z}_{i,j,...}$, forming a self-supervised learning. Two key limitations emerge: the ill-posedness of attribute prediction and the inability to capture deep quantum mechanical features. These limitations, mentioned in the introduction, can be observed in Figure 1(a). Additionally, denoising methods (Zaidi et al., 2023; Feng et al., 2023) noise the atomic positions, where the modified molecule is denoted as $\hat{S} = \{(\mathbf{z}_1, \mathbf{p}_1), ..., (\mathbf{z}_i, \mathbf{p}_i + \epsilon_1), ..., (\mathbf{z}_i, \mathbf{p}_j + \epsilon_2), ..., (\mathbf{z}_N, \mathbf{p}_N)\}$. The GNNs are required to produce equivariant features and the $\text{PRED}(\cdot)$ is used to predict noises $\epsilon_1, \epsilon_2, ...$ . These methods assume that Boltzmann distribution (i.e. exponent of PES) around equilibrium positions can be approximated by Gaussian mixture distribution (Zaidi et al., 2023). However, the assumed distribution can not always approximate the true distribution. In fact, the shapes of PES in local minima are diverse and unknown in advance (Messiah, 2014), making it challenging to define the parameters of the Gaussian mixture distribution.

### 3.2 EQUIVARIANT MASKED POSITION PREDICTION (EMPP)

The overall framework of EMPP is depicted in Figure 2. We begin the process by masking the position of an atom, causing the corresponding node in the graph to vanish. Next, we utilize equivariant backbones to generate equivariant node embeddings for the masked molecule. These embeddings are then input into a position prediction module, which outputs distributions for directions and radius. Both of them determine the predicted position. By aligning the true and predicted positions, EMPP enables the GNNs to capture atomic interactions within 3D molecular structure.

### 3.2.1 MASK POSITION AND NEIGHBOUR ENCODING

To address the limitations of previous works, EMPP only masks the atomic position and predicts it using its atomic attributes and the representations of unmasked atoms. Assuming the $i$-th atom is

masked, the modified molecule is denoted as $\hat{S} = \{(\mathbf{z}_1, \mathbf{p}_1), ..., (\mathbf{z}_i), ..., (\mathbf{z}_N, \mathbf{p}_N)\}$. According to physics, the force at equilibrium is zero, with atomic force primarily governed by atomic interactions, such as Coulombic forces (Messiah, 2014). Thus, EMPP aims to find a position in equilibrium structures that satisfies the condition $\sum_{j=1, j\neq i}^{N} \mathbf{force}_{ji} = \mathbf{0}$, where $\mathbf{force}_{ji}$ represents the inter-atomic force and is a function of the unknown masked position $\mathbf{p}_i = (x, y, z)$, given that the atomic attributes are fixed. Intuitively, there exists a unique optimal position under the force condition, since the number of unknown variables matches the number of equations. Organic chemistry further supports that the position of atoms is uniquely determined in most cases. For instance, in Figure 1, when any carbon atom is masked, its position can still be uniquely determined. Similarly, for the iodine atom, its position can be uniquely identified once the structure of its neighboring atoms is known. Further details and examples can be found in Appendix B.1. Additionally, EMPP employs equivariant representation capable of storing vector features, enabling the accurate description of the true $\mathbf{force}_{ji}$. Therefore, EMPP can be regarded as a nearly well-posed method, effectively avoiding the approximation of Gaussian mixture distributions to learn forces or other quantum features.

In the feedforward process of backbone GNN, the node of masked atom is completely removed. We encode the attributes $\mathbf{z}_i$ of the masked atom into the embedding of unmasked atoms. Note that traditional backbone GNNs use an embedding layer to project atomic attributes into the embedding space; we employ the same module augmented with a two-layer multi-layer perceptron (MLP) to embedding the attributes $\mathbf{z}_i$:

$$\mathbf{e}_i = \text{MLP}\big(\text{EMBED}(\mathbf{z}_i)\big). \tag{6}$$

Next, we aggregate $\mathbf{e}_i$ with the embeddings of the unmasked nodes:

$$\mathbf{v}_j \leftarrow \mathbf{v}_j + \mathbf{e}_i. \quad j \in \{1, 2, ...i-1, i+1, ..., N\}, \tag{7}$$

where $\mathbf{v}$ denotes the node embedding. We apply equation 7 at each layer of the backbone GNN. Notably, $\mathbf{e}_i$ is invariant under SO(3) transformation. If the node embedding $\mathbf{v}_j$ is a spherical harmonic representation, we aggregate $\mathbf{e}_i$ with the its type-0 vector $\mathbf{v}_j^{(0)}$.

Masking multiple atoms in EMPP increases the complexity of solution space, may make the position prediction ill-posed[2]. To mitigate this issue, we mask different atoms sequentially within a single molecule. First, we randomly generate the indices of the masked atoms as $Mask = \{Mask_1, Mask_2, ..., Mask_n\}$. We then create multiple masked molecules, each containing only one masked atom, denoted as $\hat{S}_1, \hat{S}_2, ..., \hat{S}_n$. Each masked molecule independently predicts the position of its masked atom, and we average the resulting loss values. The objective function for training multiple masked molecules is given by:

$$\mathcal{L}_{multi} = \frac{1}{n} \sum_i \mathcal{L}_{single}\big(\mathbf{p}_{Mask_i}, \text{Pred}(\text{GNN}(\hat{S}_i))\big), \tag{8}$$

where $\text{Pred}(\cdot)$ refers to the position prediction block, and $\mathcal{L}_{single}(\cdot)$ represents the loss function for each individual masked atom. Details of them will be provided in the following sections. Since the nodes of masked atoms in EMPP are completely deleted, there is a distinct difference between $(\hat{S}_1, \hat{S}_2, ..., \hat{S}_n)$. In contrast, previous methods do not delete nodes but perturb the features of the nodes. Intuitively, EMPP has a stronger capability to generate a vast array of diverse data.

### 3.2.2 POSITION PREDICTION

**Equivarance for predicted position.** After masking $i$-th atom and encoding its attributes into the backbone GNN, we obtain the node representations of the masked molecule. Notably, we retain the neighbor set of the masked atom, denoted as $\mathcal{N}_i$, and use the embeddings of the neighboring nodes to predict the position. For an unmasked node with embedding $\mathbf{f}_k$ (where $k \in \mathcal{N}_i$) and its position $\mathbf{p}_k$, the probability distribution of the masked atom's position is given by $p_k(\vec{\mathbf{r}}|\mathbf{f}_k)$, with the predicted position denoted as $\hat{\mathbf{p}}_i = \mathbf{p}_k + \vec{\mathbf{r}}$. This distribution $p_k$ must satisfy both normalization and non-negativity constraints: $\int_{\Omega_{xyz}} p_k(\vec{\mathbf{r}})dV = 1$ and $p_k(\vec{\mathbf{r}}) \geq 0$ where $dV = dxdydz$ is the volume element and $\Omega_{xyz}$ represents all space in Cartesian coordinates. Another challenge is ensuring that the distribution remains SO(3) transformation-equivariant, adhering to symmetry principles:

$$p_k(\vec{\mathbf{r}}|\mathbf{f}_k) = p_k(\mathbf{R}\vec{\mathbf{r}}|\mathbf{D}\mathbf{f}_k) = p_k\big(\mathbf{R}\vec{\mathbf{r}}|\text{GNN}(\mathbf{R}\hat{S})\big), \tag{9}$$

---

[2]In organic chemistry, a local system composed of two or more atoms often exhibits symmetry, allowing for multiple possible positions (See examples in Appendix B.1).

where $\mathbf{R}$ epresents any rotation matrix, and $\mathbf{D}$ is the Wigner-D matrix corresponding to the same transformation parameters. The term $\mathbf{R}\hat{S} = \{(\mathbf{z}_i, \mathbf{R}\mathbf{p}_i)|i \in \{1, ...i-1, i+1, ...N\}\}$ denotes the rotated molecular structure. Note that equation 9 disregards translation, as its effect is neutralized by the relative position prediction $\mathbf{p}_{mask} = \mathbf{p}_k + \vec{\mathbf{r}}$.

**Distribution prediction.** Recent molecular models have demonstrated that high-degree equivariant features can effectively describe the PES and forces field (Passaro & Zitnick, 2023; Liao et al., 2024). Therefore, we can directly predict the distribution of masked positions using equivariant node features. In our prediction module, we use each neighboring atom to independently predict a positional distribution, which is then aggregated. In practice, we first pass the node embeddings through an equivariant two-layer MLP. The mathematical process is expressed as:

$$\mathbf{f}_k \leftarrow \text{GATE}\big(\text{ELINEAR}(\mathbf{f}_k)\big) \tag{10}$$

$$\mathbf{f}_k \leftarrow \text{ELINEAR}\big(\mathbf{f}_k \otimes \text{EMBED}(\mathbf{z}_i)\big), \tag{11}$$

where $\text{ELINEAR}(\cdot)$ denotes the SO(3) linear layer, $\text{Gate}(\cdot)$ denotes gate activation using SiLU and $\otimes$ denotes the CG tensor product. In the second layer of the MLP, we integrate the features with the intrinsic attributes of the masked atom. We transform position prediction to Spherical coordinates and decompose it into two parts: the *radius distribution*, $p^{radius}(r)$, and the *directional distribution*, $p^{direction}(\theta, \phi)$. For the radius distribution, we uniformly partition the space into 128 intervals between 0.9 Åand 5 Å, covering the key organic molecular interaction distances. As distance is a rotation-invariant feature, we predict the radius distribution directly using the type-0 vector:

$$p_k^{radius} = \text{SoftMax}\big(\text{LINEAR}(\mathbf{f}_k^{(0)})/\tau\big), \tag{12}$$

where $\text{SoftMax}(\cdot)$ is used to normalize the distribution and $\tau$ is the temperature coefficient. For the directional distribution, we apply a Fourier transform to project the node representation onto the spherical surface, allowing us to capture features from all directions:

$$\mathcal{F}(\theta, \phi) = \sum_{l=0}^{L_{max}} \sum_{m=-l}^{l} \mathbf{f}_k^{l,m} Y_m^l(\theta, \phi), \tag{13}$$

The Fourier transform of spherical harmonics is described in detail in Appendix A.1.2. Afterward, we grid the spherical surface to obtain a finite feature matrix denoted as $\mathcal{F}_k^{grid}$. Specifically, $\mathbf{f}_k \in \mathbb{R}^{(L_{max}+1)^2 \times C}$ contains $C$ channels. After gridding, $\mathcal{F}_k^{grid} \in \mathbb{R}^{S \times C}$ retains information from all channels, where $S$ represents the grid sampling rate[3]. We then apply a shared MLP to project channel dimension into a 1D space, followed by the Softmax function to produce a normalized directional distribution.

$$p_k^{direction} = \text{SoftMax}\big(\text{MLP}(\mathcal{F}_k^{grid})/\tau\big), \tag{14}$$

It is important to note that operations within the channel dimension preserve overall equivariance. As a result, equation 14 satisfies the condition outlined in equation 9. Multi-channel features are utilized to retain high-frequency components. While EMPP's method of masking atoms and predicting coordinates using spherical harmonic projection shares similarities with Symphony (Daigavane et al., 2024), their primary focuses diverge significantly: Symphony tackles molecular generation, whereas EMPP addresses a self-supervised task. This difference in objective leads to substantial variations in implementation, such as masking strategies and the design of the multi-channel spherical harmonic projection module. Detailed comparisons are provided in Appendix B.5.

### 3.2.3 LOSS FUNCTION

**Radius loss.** During training, the model is optimized by the distribution of true radius $||\vec{\mathbf{r}}_{ik}||$ and direction $\vec{\mathbf{r}}_{ik}/||\vec{\mathbf{r}}_{ik}||$, where $\vec{\mathbf{r}}_{ik} = \mathbf{p}_i - \mathbf{p}_k$. Intuitively, the deterministic vector $\vec{\mathbf{r}}_{ik}$ should correspond to a Dirac delta distribution denoted as $\delta(\vec{\mathbf{r}} - \vec{\mathbf{r}}_{ik})$. In practice, however, the model can learn the accurate $\vec{\mathbf{r}}_{ik}$ as long as the defined distribution can uniquely represent $\vec{\mathbf{r}}_{ik}$. The Dirac delta distribution can reduce training stability and lacks the ability to transfer to other conformations. To address this, we use a Gaussian distribution[4] as a surrogate:

$$q_k^{radius}(r) \sim \mathcal{N}(r|||\vec{\mathbf{r}}_{ik}||, \sigma), \tag{15}$$

---

[3]The grid sampling rate must follow the Nyquist rate criterion: $S \geq (2L_{max})^2$.

[4]A common method to embedding distance (Thomas et al., 2018; Zitnick et al., 2022)

where $\sigma$ is typically set to 0.5 Å. The Kullback-Leibler divergence (KL-div) is utilized to compute the loss:

$$\mathcal{L}^{radius} = \frac{1}{|\mathcal{N}_i|} \sum_{k \in \mathcal{N}_i} \sum_S q_k^{radius} \log \frac{q_k^{radius}}{p_k^{radius}}. \tag{16}$$

**Direction loss.** Similarly, we choose a soft directional distribution instead of a Dirac delta function to represent the ground truth for direction:

$$w_k(\theta, \phi) = \exp\Big(\sum_{l,m} \mathrm{sh}^{l,m}(\frac{\vec{\mathbf{r}}_{ik}}{||\vec{\mathbf{r}}_{ik}||})Y_m^l(\theta, \phi)\Big), \quad q_k^{direction}(\theta, \phi) \sim \frac{1}{W} w_k(\theta, \phi), \tag{17}$$

where $\mathrm{sh}(\cdot)$ denotes the spherical harmonics representation of the direction, and $W = \int_{\Omega_{\theta\phi}} w_k(\theta, \phi) d\theta d\phi$ serves to normalize the distribution. The true distribution is then gridded, and the KL divergence between $p_k^{direction}$ and the true distribution is computed:

$$\mathcal{L}^{direction} = \frac{1}{|\mathcal{N}_i|} \sum_{k \in \mathcal{N}_i} \sum_S q_k^{direction} \log \frac{q_k^{direction}}{p_k^{direction}}. \tag{18}$$

The total loss for a single masked atom, as defined in equation 8, is given by $\mathcal{L}^{single} = \mathcal{L}^{radius} + \mathcal{L}^{direction}$.

### 3.3 APPLICATION OF EMPP

**Pre-training without annotation.** The previously described EMPP can learn quantum knowledge from equilibrium molecules without quantum property labels. Therefore, EMPP can be used to pre-train GNNs to learn transferable knowledge. For example, the PCQM4Mv2 dataset (Nakata & Shimazaki, 2017) contains a vast collection of equilibrium molecules but provides labels for only one chemical property, the gap between HOMO and LUMO. We can use EMPP to pre-train GNNs on PCQM4Mv2 and use the pre-train models to predict additional chemical properties in other datasets. **Auxiliary task for property prediction.** When training a model for a specific property prediction, we calculate two losses: the prediction loss for the targeted property and the EMPP loss as defined in equation 8. These losses are then combined for gradient descent. In this case, the goal of EMPP is not to learn the equilibrium position of the masked atom, but rather to learn the position corresponding to the known property value, which may be in a non-equilibrium state. For example, when the force on the masked atom is non-zero (indicating non-equilibrium), EMPP can capture the relationship between the true position and this force. Similarly, for other quantum labels, implicit relationships exist with the masked atom's position. In practice, we encode the label into the GNN. If the label is global invariant property like energy, the encoding can be written as:

$$\mathbf{v}_k^{(0)} \leftarrow \mathbf{v}_k^{(0)} + \mathrm{LINEAR}\big(\mathrm{GAUSS}(y^{energy*})\big), \tag{19}$$

where $\mathrm{GAUSS}(\cdot)$ denotes a Gaussian block. Specifically, for node-wise equivariant labels such as force $\mathbf{y}_i^{forces*}$, we map it into the spherical harmonic representations and add it to embeddings:

$$\mathbf{v}_k \leftarrow \mathbf{v}_k + \mathrm{ELINEAR}\big(\mathrm{sh}(\mathbf{y}_i^{forces*})\big). \tag{20}$$

Then, we use the GNN output embeddings to predict position. In this scenario, we treat EMPP as an auxiliary task for property prediction. By modeling the relationship between the target property and position, EMPP further enhances the generalization ability of the property prediction model.

## 4 RELATED WORK

**Graph self-supervised methods.** Leveraging the inherent graph structure of molecules, many graph-based self-supervised methods have the potential to train molecular models that capture transferable knowledge. For example, Hu et al. (2020) proposed graph context prediction and attribute masking methods to enhance molecular property prediction. GraphMAE (Hou et al., 2022) pre-trained molecular models using a generative decoder to reconstruct atomic and bond attributes. D-SLA (Kim et al., 2022) applied contrastive learning based on graph edit distance, improving predictions of molecular biochemical activities. Additionally, graph motifs—induced subgraphs that describe recurrence and significance—have increasingly been utilized to construct self-supervised learning frameworks for molecules (Rong et al., 2020; Zhang et al., 2021; Inae et al., 2023), facilitating the learning of multi-scale molecular information. These pre-training methods primarily focus on graph

characteristics, while neglecting the intrinsic quantum mechanisms within molecules. As a result, they are limited in their ability to predict molecular quantum mechanical properties.

**3D molecular representation.** Given the strong correlation between the quantum characteristics of molecules and their 3D structures, recent molecular models have increasingly focused on 3D representations (Liao & Smidt, 2023; Passaro & Zitnick, 2023; Liao et al., 2024). As a result, self-supervised techniques have also evolved to operate in 3D space. For instance, Unimols (Zhou et al., 2023; Lu et al., 2023) masked atomic properties and restored them using 3D molecular models, while denoising methods (Zaidi et al., 2023; Feng et al., 2023) introduced a series of physics-informed pre-training paradigms. We provide a detailed discussion of these methods in Section 3.1 and highlight their main limitation: accurately defining the parameters of Gaussian mixture distributions can be challenging. In contrast, EMPP learns quantum mechanical features through a position prediction process, effectively bypassing the difficulties associated with denoising methods.

**Spherical harmonics projection.** High-degree spherical harmonic representations with grid projections (Liao & Smidt, 2023; Passaro & Zitnick, 2023) have demonstrated significant capabilities in spatial description. Building on this, Symphony (Daigavane et al., 2024) introduced a neighbor-based spatial position prediction method, developing a framework for molecular generation. While EMPP shares some technical similarities with Symphony, their distinct tasks (molecular generation versus self-supervised learning) lead to differing implementation priorities. Symphony prioritizes flexibility in position prediction to ensure the sampling of diverse molecules, whereas EMPP emphasizes accuracy and well-posed position prediction to facilitate learning of genuine physical interactions.

## 5 EXPERIMENTS

In this section, we present experiments to evaluate the effectiveness of EMPP across several 3D molecular benchmarks. Since EMPP can be applied in both unlabeled and labeled scenarios, we evaluate it in two settings: (i) self-supervised tasks for learning transferable molecular knowledge, and (ii) auxiliary tasks for enhancing the prediction of supervised molecular properties.

### 5.1 DATASETS AND CONFIGURATIONS

**Datasets.** We evaluate quantum property prediction using the QM9 (Ramakrishnan et al., 2014) and MD17 (Chmiela et al., 2017) datasets. QM9 comprises 134,000 stable small organic molecules made up of C, H, O, N, and F atoms, with one conformation per molecule, and includes labels for 12 quantum properties. MD17 contains molecular dynamics trajectories for 8 small organic molecules, providing between 150,000 and nearly 1 million conformations per molecule, along with corresponding total energy and force labels. Notably, MD17 features a significant number of non-equilibrium molecules. Additionally, we utilize the PCQM4Mv2 (Nakata & Shimazaki, 2017) dataset to pre-train GNN backbones, which consists of 3.4 million organic molecules, each with one equilibrium conformation, and is widely used for pre-training.

**Baselines.** Our baselines include state-of-the-art self-supervised methods for 3D molecular structures, such as AttrMask (Hu et al., 2020), DP-TorchMD-NET (Zaidi et al., 2023), 3D-EMGP (Jiao et al., 2023), SE(3)-DDM (Liu et al., 2022), Transformer-M (Luo et al., 2022), and Frad (Feng et al., 2023). These methods pre-train GNNs on the PCQM4Mv2 dataset before predicting molecular properties in QM9 and MD17, leveraging additional molecular data. In contrast, EMPP can operate without extra data, so we also compare it to molecular models trained solely on QM9 or MD17, including SchNet (Schütt et al., 2018), PaiNN (Schütt et al., 2021), DimeNet++ (Gasteiger et al., 2020), TorchMD-NET (Doerr et al., 2021), SEGNN (Brandstetter et al., 2022), and Equiformer (Liao & Smidt, 2023). Detailed configurations for EMPP and the aforementioned baselines can be found in Appendix C.1.

### 5.2 RESULTS WITHOUT PRE-TRAINING

**QM9.** We first evaluate EMPP without incorporating additional data, treating it as an auxiliary task for QM9 and MD17. The total loss comprises equation 8 combined with the MAE loss from the original property prediction. In these experiments, EMPP is implemented using the Equiformer backbone (Liao & Smidt, 2023) due to its high-degree equivariant representation, which effectively captures interatomic features. As shown in Table 1, two key observations emerge. First, EMPP enhances prediction accuracy across all QM9 tasks, achieving the best results in 11 tasks without additional data. Moreover, multi-masking show better performance, demonstrating that the data generated by EMPP is diverse and reliable. Second, while EMPP is designed to learn interaction

Table 1: Results on QM9 property prediction without pre-training on extra molecular data. † denotes using different data partitions. In this experiment, EMPP uses the Equiformer backbone. n-Mask denotes masking n atoms in each molecule during training. The masking strategy follow equation 8.

| Task | $\alpha$ | $\Delta\epsilon$ | $\varepsilon_{HOMO}$ | $\varepsilon_{LUMO}$ | $\mu$ | $C_v$ | $G$ | $H$ | $R^2$ | $U$ | $U_0$ | ZPVE |
| Units | bohr$^3$ | meV | meV | meV | D | cal/(mol K) | meV | meV | bohr$^3$ | meV | meV | meV |
|---|---|---|---|---|---|---|---|---|---|---|---|---|
| SchNet | .235 | 63 | 41 | 34 | .033 | .033 | 14 | 14 | .073 | 19 | 14 | 1.70 |
| DimeNet++ | .044 | 33 | 25 | 20 | .030 | .023 | 8 | 7 | .331 | 6 | 6 | 1.21 |
| PaiNN | .045 | 46 | 28 | 20 | .012 | .024 | 7.35 | 5.98 | .066 | 5.83 | 5.85 | 1.28 |
| TorchMD-NET | .059 | 36 | 20 | 18 | .011 | .026 | 7.62 | 6.16 | **.033** | 6.38 | 6.15 | 1.84 |
| SEGNN† | .060 | 42 | 24 | 21 | .023 | .031 | 15 | 16 | .660 | 13 | 15 | 1.62 |
| Equiformer | .046 | 30 | 15 | 14 | .011 | .023 | 7.63 | 6.63 | .251 | 6.74 | 6.59 | 1.26 |
| EMPP (1-Mask) | .041 | 27 | 14 | 13 | .0108 | .021 | 6.89 | 5.38 | .189 | 6.05 | 5.88 | 1.20 |
| EMPP (3-Mask) | **.039** | **26** | **13** | **12** | **.0096** | **.019** | **6.32** | **5.02** | .154 | **5.72** | **5.25** | **1.18** |

Table 2: Results on MD17 testing set without pre-trainging on extra molecular data. Energy and force are in units of meV and meV/ Å. The "energy only" and "force only" are based on "1-Mask" strategy.

| Methods | Aspirin | | Benzene | | Ethanol | | Malonaldehyde | | Naphthalene | | Salicylic acid | | Toluene | | Uracil | |
| | energy | forces | energy | forces | energy | forces | energy | forces | energy | forces | energy | forces | energy | forces | energy | forces |
|---|---|---|---|---|---|---|---|---|---|---|---|---|---|---|---|---|
| SchNet | 16.0 | 58.5 | 3.5 | 13.4 | 3.5 | 16.9 | 5.6 | 28.6 | 6.9 | 25.2 | 8.7 | 36.9 | 5.2 | 24.7 | 6.1 | 24.3 |
| DimeNet | 8.8 | 21.6 | 3.4 | 8.1 | 2.8 | 10.0 | 4.5 | 16.6 | 5.8 | 16.2 | 4.4 | 9.4 | 5.0 | 13.1 |
| PaiNN | 6.9 | 14.7 | - | - | 2.7 | 9.7 | 3.9 | 13.8 | 5.0 | 3.3 | 4.9 | 8.5 | 4.1 | 4.1 | 4.5 | 6.0 |
| TorchMD-NET | 5.3 | 11.0 | 2.5 | 8.5 | 2.3 | 4.7 | 3.7 | 7.3 | 3.7 | 2.6 | **4.0** | 5.6 | 3.2 | 2.9 | 4.1 | 4.1 |
| Equiformer | 5.3 | 6.6 | 2.5 | 8.1 | 2.2 | 2.9 | 3.2 | 5.4 | 4.4 | 2.0 | 4.3 | 3.9 | 3.7 | 2.1 | 4.3 | 3.4 |
| EMPP (1-Mask) | 5.1 | 6.4 | 2.3 | 7.5 | 2.2 | 2.7 | 3.0 | 5.2 | 4.1 | 1.9 | 4.3 | 3.7 | 3.5 | 2.1 | 4.1 | 3.4 |
| EMPP (3-Mask) | 5.0 | **6.2** | **2.1** | 7.3 | **2.0** | 2.6 | 3.0 | 5.0 | 3.7 | 1.6 | 4.1 | **3.6** | 3.3 | 2.0 | 4.0 | **3.0** |
| EMPP (energy only) | **4.8** | 6.6 | **2.1** | 8.5 | **2.0** | 2.5 | **2.8** | 5.1 | **3.5** | 1.5 | 4.1 | 3.9 | **3.3** | 2.2 | 4.0 | 3.2 |
| EMPP (force only) | 5.3 | 6.4 | 2.8 | **7.1** | 2.3 | **2.4** | 3.3 | **4.9** | 4.4 | 1.5 | 4.3 | 3.7 | 3.7 | **2.0** | 4.7 | 3.2 |

between adjacent atoms, it also leads to improvements in properties not directly related to atomic interaction, such as $\varepsilon_{HOMO}$ and $\varepsilon_{LUMO}$ (Pope, 1999), further confirming its effectiveness.

**MD17.** The MD17 dataset provides both global labels (energy) and node-wise labels (forces), along with numerous non-equilibrium conformations that present new challenges for models and self-supervised methods. To address this, we encode both the global energy and the forces of the masked atom into the backbone GNNs. EMPP predicts atom positions where the predicted forces align with the ground truth. The results, presented in Table 2, show that EMPP achieves the best performance on most tasks. Additionally, we conducted two experiments where either energy or force alone is encoded into the backbone, referred to as "energy only" and "force only". Notably, when only energy is encoded, Table 2 indicates that energy prediction can be further improved. In the cases of Aspirin, Naphthalene, and Toluene, the "energy only" approach with single-atom masking even surpasses the results obtained by three-atoms masking while encoding both energy and forces. These findings suggest that encoding a single label can simplify EMPP's learning process, thereby enhancing its effectiveness for predicting specific properties. We further investigate the performance of EMPP without pre-training on the GEOM-Drug dataset (Axelrod & Gomez-Bombarelli, 2022), which contains abundant non-equilibrium data. Results can be found in the Appendix C.2.

## 5.3 RESULTS WITH PRE-TRAINING ON PCQM4Mv2

In this section, we assess the effectiveness of EMPP as a pre-training task. To maintain consistency with state-of-the-art self-supervised methods (Zaidi et al., 2023; Feng et al., 2023), we use TorchMD-Net (ET) (Thölke & Fabritiis, 2022) as the backbone model. To enhance the fine granularity of the spherical representation after Fourier transformation, we extend the node representation of TorchMD-Net to $L_{max} = 3$ while retaining all other core operations (see details in Appendix B.3). During the pre-training phase, EMPP encodes only the atomic numbers of the masked atoms into the backbone. In the fine-tuning stage, EMPP encodes properties following the same approach outlined in Section 5.2.

As shown in Table 3, the attribute masking method produces poorer results for quantum property prediction, consistent with our earlier observations: recovering attributes based on simple features limits the model's ability to capture deep quantum characteristics. In contrast, denoising methods such as DP-TorchMD-Net and Frad deliver competitive performance due to their physically interpretable paradigms. Notably, our EMPP surpasses denoising methods in nine tasks, achieving the best results in seven tasks. This success is attributed to EMPP's more precise paradigm, which employs a precise paradigm to learn interactions instead of approximation of mixture distribution.

Table 3: Results on QM9 property prediction with pre-trainging on extra molecular data. EMPP use the TorchMD-Net backbone. Bold and underline indicate the best result, and the second best result.

| Task | $\alpha$ | $\Delta\epsilon$ | $\varepsilon_{HOMO}$ | $\varepsilon_{LUMO}$ | $\mu$ | $C_v$ | $G$ | $H$ | $R^2$ | $U$ | $U_0$ | ZPVE |
|------|----------|------------------|----------------------|----------------------|-------|-------|-----|-----|-------|-----|-------|------|
| Units | bohr$^3$ | meV | meV | meV | D | cal/(mol K) | meV | meV | bohr$^3$ | meV | meV | meV |
| AttrMask | .072 | 50.0 | 31.3 | 37.8 | .020 | 0.062 | 11.2 | 11.4 | .423 | 10.8 | 10.7 | 1.90 |
| Transformer-M | .041 | 27.4 | 17.5 | 16.2 | .037 | 0.022 | 9.63 | 9.39 | **.075** | 9.41 | 9.37 | **1.18** |
| SE(3)-DDM | .046 | 40.2 | 23.5 | 19.5 | .015 | .024 | 7.65 | 7.09 | .122 | 6.99 | 6.92 | 1.31 |
| 3D-EMGP | .057 | 37.1 | 21.3 | 18.2 | .020 | .026 | 9.30 | 8.70 | .092 | 8.60 | 8.60 | 1.38 |
| DP-TorchMD-Net | .0517 | 31.8 | 17.7 | 14.3 | .012 | .020 | 6.91 | 6.45 | .4496 | 6.11 | 6.57 | 1.71 |
| Frad | 0.037 | 27.8 | 15.3 | 13.7 | **.010** | .020 | **6.19** | **5.55** | .342 | 5.62 | 5.33 | 1.42 |
| EMPP (3-Mask) | **.035** | **25.8** | **13.7** | **13.4** | .014 | **.019** | 6.45 | 5.73 | .241 | **5.34** | **5.08** | 1.27 |

Table 4: Ablation study on key hyperparameters. Bold indicates the default configuration.

| Sampling Rate | $\alpha$ | $\varepsilon_{HOMO}$ | Loss Weight | $\alpha$ | $\varepsilon_{HOMO}$ |
|---------------|----------|----------------------|-------------|----------|----------------------|
| $20^2$ | .050 | 20.4 | 0.1 | .046 | 15.4 |
| $50^2$ | .043 | 18.5 | 0.5 | .045 | 15.2 |
| **$100^2$** | .041 | 14.2 | **1** | .041 | 14.2 |
| $150^2$ | .041 | 14.7 | 5 | .044 | 15.8 |

## 5.4 ABLATION STUDY

In this section, we address several key questions: (i) the relationship between parameters of Gaussian mixtures and performance in denoising method. (ii) the impact of key hyper-parameters in EMPP. Additional ablation studies are provided in Appendix C.2.

To assess the impact of Gaussian mixture distributions in denoising, we applied the DP-TorchMD-Net (Zaidi et al., 2023) to QM9 (HOMO, LUMO) as an auxiliary task, with the results displayed in Figure 3. By varying the standard deviation $\sigma$ of the Gaussian mixture, which controls the curvature, we observed changes in the model's generalization performance. The experiment with $\sigma = 0$ is equivalent to training without denoising. From Figure 3, we derive two key insights: (i) Without utilizing external data, the denoising method demonstrates limited potential to improve the model's generalization ability, yielding results that are similar to or worse than the baseline without denoising. (ii) The performance of denoising is highly sensitive to the standard deviation, with different values leading to significantly divergent outcomes. In practical applications, because the curvature near local minima is often unknown,

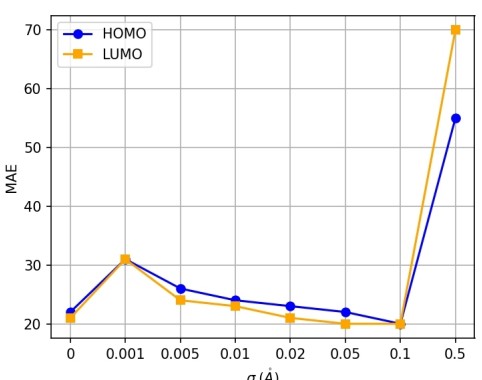

Figure 3: The curve of performance varying with the standard deviation $\sigma$.

the optimal standard deviation typically requires empirical tuning, making it challenging to account for all possible local minima.

Another ablation experiment is to verify the key hyperparameters in EMPP, including the sampling rate and loss weight. From the Table 4, we draw two conclusions: (i) A high sampling rate over the sphere is beneficial for better predicting all possible positions, thereby enhancing generalization ability. (ii) When EMPP is used as an auxiliary task, it can be of the same importance as the original task.

## 6 CONCLUSION

We identified key limitations in mainstream molecular self-supervised learning: attribute masking introduces ill-posedness and fails to capture quantum features, while denoising struggles with accurately modeling Gaussian mixture in unknown distributions. To address these issues, we propose EMPP, which predicts masked atom positions through neighbor structures, bypassing Gaussian mixture approximation and turning the task into a well-posed one. Our experiments show EMPP achieves state-of-the-art results in quantum property prediction. While we employ equivariant representations in EMPP, extending to higher-order representation ($l > 3$) remains an open question.

## ACKNOWLEDGEMENTS

We are thankful to the anonymous reviewers for their helpful comments. The computations in this research were performed using the CFFF platform of Fudan University. This research was supported by the National Natural Science Foundation of China under Grant 82394432 and 92249302. The corresponding authors are Fenglei Cao and Yuan Qi.

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

APPENDIX

## A  THE MATHEMATICS

### A.1  THE MATHEMATICS OF SPHERICAL HARMONICS

#### A.1.1  THE PROPERTIES OF SPHERICAL HARMONICS

The spherical harmonics $Y_l^m(\theta, \phi)$ are the angular portion of the solution to Laplace's equation in spherical coordinates where azimuthal symmetry is not present. Some care must be taken in identifying the notational convention being used. In this entry, $\theta$ is taken as the polar (colatitudinal) coordinate with $\theta$ in $[0, \pi]$, and $\phi$ as the azimuthal (longitudinal) coordinate with $\phi$ in $[0, 2\pi]$.

Spherical harmonics satisfy the spherical harmonic differential equation, which is given by the angular part of Laplace's equation in spherical coordinates. If we define the solution of Laplace's equation as $F = \Phi(\phi)\Theta(\theta)$, the equation can be transformed as:

$$\frac{\Phi(\phi)}{\sin\theta}\frac{d}{d\theta}\left(\sin\theta\frac{d\Theta}{d\theta}\right) + \frac{\Theta(\theta)}{\sin^2\theta}\frac{d^2\Phi(\phi)}{d\phi^2} + l(l+1)\Theta(\theta)\Phi(\phi) = 0 \tag{21}$$

Here we omit the derivation process and just show the result. The (complex-value) spherical harmonics are defined by:

$$Y_m^l(\theta, \phi) \equiv \sqrt{\frac{2l+1}{4\pi}\frac{(l-m)!}{(l+m)!}}P_m^l(\cos\theta)e^{im\phi}, \tag{22}$$

where $P_m^l(\cos\theta)$ is an associated Legendre polynomial. Spherical harmonics are integral basis, which satisfy:

$$\begin{aligned}&\int_0^{2\pi}\int_0^{\pi}Y_{m_1}^{l_1}(\theta,\phi)Y_{m_2}^{l_2}(\theta,\phi)Y_{m_3}^{l_3}(\theta,\phi)\sin\theta\,d\theta\,d\phi\\&=\sqrt{\frac{(2l_1+1)(2l_2+1)(2l_3+1)}{4\pi}}\begin{pmatrix}l_1 & l_2 & l_3\\0 & 0 & 0\end{pmatrix}\begin{pmatrix}l_1 & l_2 & l_3\\m_1 & m_2 & m_3\end{pmatrix},\end{aligned} \tag{23}$$

where $\begin{pmatrix} l_1 & l_2 & l_3 \\ m_1 & m_2 & m_3 \end{pmatrix}$ is a Wigner 3j-symbol (which is related to the Clebsch-Gordan coefficients). We list a few spherical harmonics which are:

$$Y_0^0(\theta, \phi) = \frac{1}{2}\sqrt{\frac{1}{\pi}}$$

$$Y_{-1}^1(\theta, \phi) = \frac{1}{2}\sqrt{\frac{3}{2\pi}}\sin\theta e^{-i\phi}$$

$$Y_0^1(\theta, \phi) = \frac{1}{2}\sqrt{\frac{3}{\pi}}\cos\theta$$

$$Y_1^1(\theta, \phi) = \frac{-1}{2}\sqrt{\frac{3}{2\pi}}\sin\theta e^{i\phi}$$

$$Y_{-2}^2(\theta, \phi) = \frac{1}{4}\sqrt{\frac{15}{2\pi}}\sin^2\theta e^{-2i\phi} \quad (24)$$

$$Y_{-1}^2(\theta, \phi) = \frac{1}{2}\sqrt{\frac{15}{2\pi}}\sin\theta\cos\theta e^{-i\phi}$$

$$Y_0^2(\theta, \phi) = \frac{1}{4}\sqrt{\frac{5}{\pi}}\left(3\cos^2\theta - 1\right)$$

$$Y_1^2(\theta, \phi) = \frac{-1}{2}\sqrt{\frac{15}{2\pi}}\sin\theta\cos\theta e^{i\phi}$$

$$Y_2^2(\theta, \phi) = \frac{1}{4}\sqrt{\frac{15}{2\pi}}\sin^2\theta e^{2i\phi}$$

In this work, we use the real-value spherical harmonics rather than the complex-value one.

### A.1.2 FOURIER TRANSFORMATION OVER $S^2$

It is well known that the spherical harmonic $Y_m^l$ form a complete set of orthonormal functions and thus form an orthonormal basis of the Hilbert space of square-integrable function. On the unit sphere $S^2$, any square-integrable function $f$ can thus be expanded as a linear combination of these:

$$f(\theta, \phi) = \sum_{l=0}^{\infty}\sum_{m=-l}^{l} f_m^l Y_m^l(\theta, \phi), \quad (25)$$

The coefficient $f_m^l$ can be obtained by the Fourier transformation over $S^2$, which is

$$f_m^l = \int_{S^2} f(\vec{r})Y_m^{l*}(\vec{r})d\vec{r} = \int_0^{2\pi}\int_0^{\pi} d\theta\sin\theta f(\theta, \psi)Y_m^{l*}(\theta, \psi). \quad (26)$$

Usually we define a vector $\mathbf{f}^l = [f_{-l}^l, f_{-l+1}^l, ..., f_l^l]$ to denote the Fourier coefficients with degree $l$. We now investigate how the fourier coefficients transforms if we rotate the input signal. More precisely, we want to calculate the coefficient $\mathbf{f}_{\mathbf{R}}^l$ of the signal $f(\mathbf{R}\vec{r})$, where $\mathbf{R} \in SO(3)$ is a rotation matrix.

Using the fact $\mathbf{Y}^l(\mathbf{R}\vec{r}) = \mathbf{D}^l(\mathbf{R})\mathbf{Y}^l(\vec{r})$, and equation 25, we know

$$f(\mathbf{R}\vec{r}) = \sum_{l=0}^{\infty}\sum_{m=-l}^{l} f_m^l Y_m^l(\mathbf{R}\vec{r}) = \sum_{l=0}^{\infty}\sum_{m=-l}^{l} f_m^l \sum_{m'} \mathbf{D}_{mm'} Y_{m'}^l(\vec{r}).$$

Therefore $\mathbf{f}_{\mathbf{R}}^l = \mathbf{D}^T\mathbf{f}^l$ and it is steerable.

### A.1.3 THE RELATIONSHIP BETWEEN SPHERICAL HARMONICS AND WIGNER-D MATRIX

A rotation $\mathbf{R}$ sending the $\vec{r}$ to $\vec{r}'$ can be regarded as a linear combination of spherical harmonics that are set to the same degree. The coefficients of linear combination represent the complex conjugate of an element of the Wigner D-matrix. The rotational behavior of the spherical harmonics is perhaps

their quintessential feature from the viewpoint of group theory. The spherical harmonics $Y_m^l$ provide a basis set of functions for the irreducible representation of the group SO(3) with dimension $(2l+1)$.

The Wigner-D matrix can be constructed by spherical harmonics. Consider a transformation $Y_m^l(\vec{r}) = Y_m^l(\mathbf{R}_{\alpha,\beta,\gamma}\vec{r}_x)$, where $\vec{r}_x$ denote the x-orientation. $\alpha, \beta, \gamma$ denotes the items of Euler angle. Therefore, $Y_m^l(\vec{r})$ is invariant with respect to rotation angle $\gamma$. Based on this viewpoint, the Wigner-D matrix with shape $(2l+1) \times (2l+1)$ can be defined by:

$$D_m^l(\mathbf{R}_{\alpha,\beta,\gamma}) = \sqrt{2l+1}Y_m^l(\vec{r}). \tag{27}$$

In this case, the orientations are encoded in spherical harmonics and their Wigner-D matrices, which are utilized in our cross module.

## A.2 EQUIVARIANT OPERATION

### A.2.1 EQUIVARIANCE OF CLEBSCH-GORDAN TENSOR PRODUCT

The Clebsch-Gordan Tensor Product shows a strict equivariance for different group representations, which make the mixture representations transformed equivariant based on Wigner-D matrices. We use $D_{m_1', m_1}$ to denote the element of Wigner-D matrix. The Clebsch-Gordan coefficient satisfies:

$$
\begin{aligned}
& \sum_{m_1', m_2'} C_{(l_1, m_1')(l_2, m_2')}^{(l_0, m_0)} D_{m_1' m_1}^{l_1}(g) D_{m_2' m_2}^{l_2}(g) \\
& = \sum_{m_0'} D_{m_0 m_0'}^{l_0}(g) C_{(l_1, m_1)(l_2, m_2)}^{(l_0, m_0')}
\end{aligned}
\tag{28}
$$

Therefore, the spherical harmonics can be combined equivariantly by CG Tensor Product:

$$
\begin{aligned}
& CG \left( \sum_{m_1'} D_{m_1 m_1'}^{l_1}(g) Y_{m_1'}^{l_1}, \sum_{m_2'} D_{m_2 m_2'}^{l_2}(g) Y_{m_2'}^{l_2} \right)_{m_0}^{l_0} \\
& = \sum_{m_1, m_2} C_{(l_1, m_1)(l_2, m_2)}^{(l_0, m_0)} \sum_{m_1'} D_{m_1 m_1'}^{l_1}(g) Y_{m_1'}^{l_1} \sum_{m_2'} D_{m_2 m_2'}^{l_2}(g) Y_{m_2'}^{l_2} \\
& = \sum_{m_0'} D_{m_0 m_0'}^{l_o}(g) \sum_{m_1, m_2} C_{(l_1, m_1)(l_2, m_2)}^{(l_0, m_0')} Y_{m_1'}^{l_1} Y_{m_2'}^{l_2} \\
& = \sum_{m_0'} D_{m_0 m_0'}^{l_0}(g) CG_{m_0'}^{l_o} \left( Y_{m_1'}^{l_1}, Y_{m_2'}^{l_2}, \right).
\end{aligned}
\tag{29}
$$

equation 29 represents a relationship between scalar. If we transform the scalar to vector or matrix, equation 29 is equal to

$$(\mathbf{D}_\mathbf{R}^{l_1}\mathbf{u} \otimes \mathbf{D}_\mathbf{R}^{l_2}\mathbf{v})^l = \mathbf{D}_\mathbf{R}^l(\mathbf{u} \otimes \mathbf{v})^l. \tag{30}$$

The tensor CG product mixes two representations to a new representation under special rule. For example, 1.two type-0 vectors will only generate a type-0 representations; 2.type-$l_1$ and type-$l_2$ can generate type-$l_1 + l_2$ vector at most. Note that some widely-used products are related to tensor product: scalar product ($l_1 = 0$, $l_2 = 1$, $l = 1$), dot product ($l_1 = 1$, $l_2 = 1$, $l = 0$) and cross product ($l_1 = 1$, $l_2 = 1$, $l = 1$). However, for each element with $l > 0$, there are multi mathematical operation for the connection with weights. The relation between number of operations and degree is quadratic. Thus, as degree increases, the amount of computation increases significantly, making calculation of the CG tensor product slow for higher order irreps. This statement can be proven by the implementation of e3nn (o3.FullyConnectedTensorProduct).

### A.2.2 LEARNABLE PARAMETERS IN TENSOR PRODUCT

We utilize the e3nn library (Geiger et al., 2022) to implement the corresponding tensor product. It is crucial to emphasize that the formulation of CG tensor product is devoid of any learnable parameters, as CG coefficients remain constant. In the context of e3nn, learnable parameters are introduced into each path, represented as $w(\mathbf{u}^{l_1} \otimes \mathbf{v}^{l_2})$. Importantly, these learnable parameters will not destory the equivariance of each path. However, they are limited in capturing directional information. In equivariant models, the original CG tensor product primarily captures directional information. We have previously mentioned our replacement of the CG tensor product with learnable modules. It is worth noting that our focus lies on the CG coefficients rather than the learnable parameters in the e3nn implementation.

### A.2.3 GATE ACTIVATION AND NORMALIZATION

The gate activation and normalization used in HDGNN are implement by e3nn code framework.

**Gate Activation.** In equivariant models, the gate activation combines two sets of group representations. The first set consists of scalar irreps ($l = 0$), which are passed through standard activation functions such as sigmoid, ReLU and SiLU. The second set comprises higher-order irreps (($l > 0$)), which are multiplied by an additional set of scalar irreps that are introduced solely for the purpose of the activation layer. These scalar irreps are also passed through activation functions.

The gate activation allows for the controlled integration of different types of irreps in the network. The scalar irreps capture global and local patterns, while the higher-order irreps capture more complex relationships and interactions. By combining these irreps in a gate-like manner, the gate activation enables the model to selectively amplify or suppress information flow based on the importance of different irreps for a given task.

**Normalization.** Normalization is a technique commonly used in neural networks to normalize the activations within each layer. It helps stabilize and accelerate the training process by reducing the internal covariate shift, which refers to the change in the distribution of layer inputs during training.

The normalization process involves computing the mean and variance across the channels. In equivariant normalization, the variance is computed using the root mean square value of the L2-norm of each type-$l$ vector. Additionally, this normalization removes the mean term. The normalized activations are then passed through a learnable affine transformation without a learnable bias, which enables the network to adjust the mean and variance based on the specific task requirements.

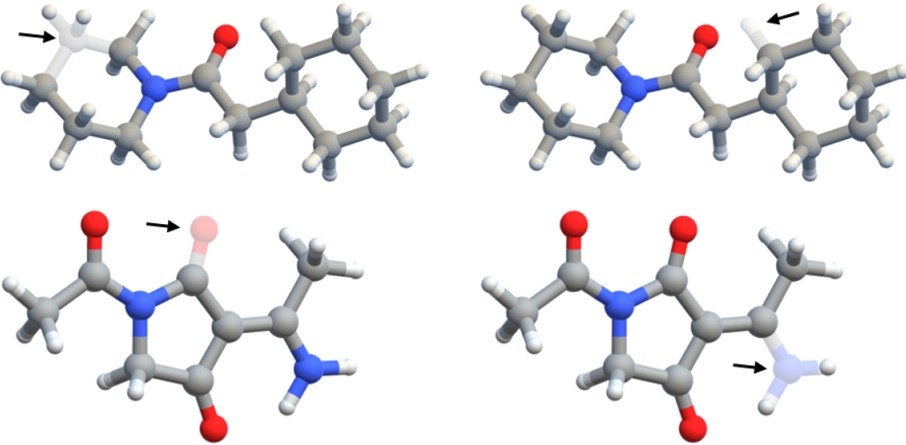

Figure 4: Determinacy of atomic positions in organic molecules. The relationship between atoms and colors is (Carbon, gray; Hydrogen, white; Oxygen, red; Nitrogen, blue). When the system is fixed, most atoms have a uniquely determined position. The translucent area represents the atom of interest, and the arrow indicates its unique position.

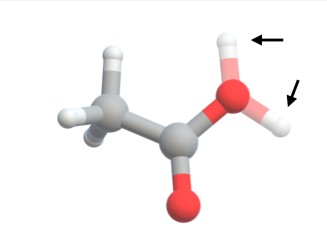

Figure 5: Atoms with uncertain positions exist. In organic molecules, the positions of some atoms have multiple possibilities, corresponding to several local minima of the potential energy surface. We represent the two possible positions of the atom with arrows.

## B    Model Details

### B.1    The Position Distribution in Organic Chemistry

In the realm of organic chemistry, the majority of atomic positions within molecules are unequivocally determined, reflecting the precise architectural principles that govern molecular structure. This certainty stems from the well-defined rules of covalent bonding, hybridization, and the tetrahedral geometry that (C, H, O, N, F, S, ...) atoms often exhibit, which together constrain the spatial arrangement of atoms in most organic compounds. As illustrated in the Figure 4, for a given molecule, the majority of its constituent atoms possess uniquely determined positions. Any other arrangements would result in an implausible molecular structure.

However, there exists a minority subset of scenarios where atomic positions may be ambiguous due to the existence of multiple potential locations, corresponding to various local minima on the potential energy surface. These sites of uncertainty are typically associated with conformational flexibility, such as in molecules with large rotational freedom about single bonds, or in cases where there are multiple stable conformations or rotameric states. We take Figure 5 as example, the hydroxyl group of the molecule can rotate, allowing for two possible positions of the hydrogen atom.

Furthermore, when considering the positional possibilities of two atoms simultaneously, the combinatorial complexity of their potential configurations increases significantly. This multiplicative effect is a consequence of the interplay between intramolecular forces and the three-dimensional nature of molecular space, which allows for a vast array of spatial orientations and relative positions. As illustrated in the Figure 6, masking multiple atoms introduces a multitude of possibilities, rendering the position prediction of EMPP an ill-posed problem.

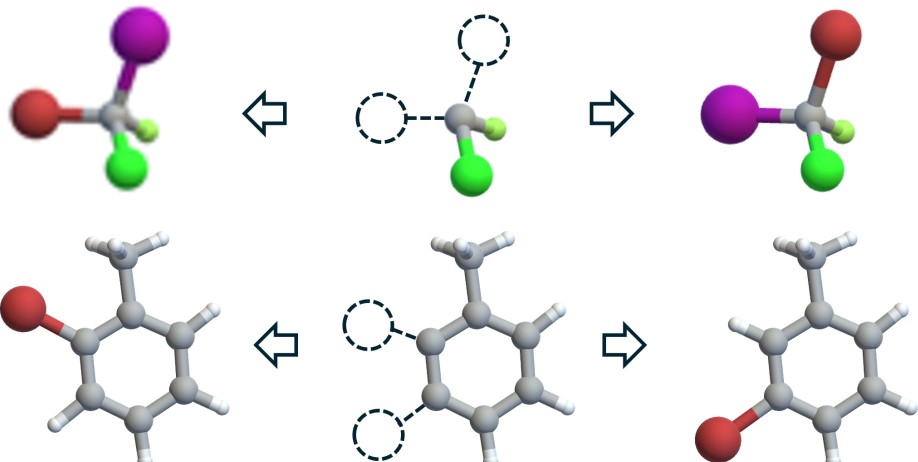

Figure 6: The Possible positions of multiple atoms are masked. When calculating the forces acting on each atom, the influence of another masked atom must also be taken into account.

In summary, while most atomic positions in organic molecules are rigidly defined, there are instances where positional ambiguity arises, and this complexity is magnified when multiple atoms are considered. This interplay between certainty and uncertainty in atomic positioning is a fascinating aspect of organic chemistry that has profound implications for our understanding of molecular structure and behavior.

### B.2    Details of Prediction Module

Our position prediction module consists of multiple equivariant operations, designed based on the structure of the backbone. For the Equiformer, the output feature $\mathbf{f}_k$ is represented as "128x0e+64x1e+32x2e" (irreducible representations in e3nn (Geiger et al., 2022)). We first standardize the number of channels using a linear layer, resulting in "64x0e+64x1e+64x2e". Through a gating mechanism and a second linear layer, $\mathbf{f}_k$ is mapped to a representation of "32x0e+32x1e+32x2e".

Following Fourier transformation and gridding, we apply a shared MLP on the spherical surface, which is equivalent to a 1D convolution. Since nonlinear transformations on the sphere do not disrupt equivariance, the shared MLP first projects the channels into a 16-dimensional space and then applies the SiLU activation function. Finally, the 16-dimensional vector is projected back down to one dimension, yielding a non-normalized distribution.

## B.3 TORCHMD-NET WITH HIGHER-ORDER SPHERICAL HARMONIC REPRESENTATION

In the position prediction module, the EMPP must map the equivariant representations onto a distribution on the spherical surface. A greater diversity of frequencies in the frequency domain representation leads to a finer time-domain representation post-transformation. EMPP is built upon Equiformer and TorchMD-Net; while Equiformer incorporates high-order spherical harmonics representations (analogous to higher frequencies), TorchMD-Net includes only invariant features of $l = 0$ and three-dimensional equivariant features of $l = 1$. Consequently, the output after Fourier transformation may not effectively capture the fine-grained positional distribution. To address this, we convert the TorchMD-Net embeddings into high-order spherical harmonics representations. First, we project the embeddings onto representations of $l = [0, 1, 2, 3]$ using the spherical harmonics function. Next, we expand the number of attention coefficients from TorchMD-Net, applying them to each degree of representation (whereas the original model only applied these coefficients to the $l = 1$ representation). When calculating the inner product, the inner product of high-order spherical harmonics remains rotationally invariant, ensuring that the modified TorchMD-Net maintains overall equivariance. We apply this variant in our experiments. Note that we do not change the core operations in TorchMD-Net.

## B.4 PREDICTION WITH NEIGHBORS

In equations equation 16 and equation 18, we restrict the prediction of masked positions to neighboring nodes within the cutoff radius. This restriction arises from the fact that, in GNNs, atoms beyond the cutoff do not directly interact with the masked atom; instead, they influence the PES indirectly by transmitting features to the embeddings of neighboring nodes through multi-layer forward propagation. Including points outside the cutoff during loss calculation would compromise the inductive bias inherent in GNNs. Furthermore, our ablation study demonstrates that incorporating atoms beyond the cutoff can diminish the model's generalization performance.

## B.5 DIFFERENCES FROM SYMPHONY

Symphony (Daigavane et al., 2024) proposes a molecular generation framework based on equivariant representations. It constructs a fragment sequence by iteratively masking atoms, with each subsequent fragment masking one additional atom. Symphony then uses spherical harmonic projections of high-degree equivariant representations to encode the spatial distribution of atomic positions, enabling it to sample new atoms based on the predicted distribution. EMPP uses the similar technique to define position prediction module where the relative positions are projected onto a spatial distribution based on high-degree equivariant representations. However, EMPP and Symphony differ in several key aspects:

- **Masking Strategy:** Symphony, geared towards molecular generation, masks all information about an atom, relying on the model's prediction of a "focus" atom to sample subsequent atoms. EMPP, conversely, masks only the position of a single atom, retaining the ground truth of its atomic type and neighbor information. EMPP's position inference leverages all available real information from the molecule. These distinct strategies reflect their respective goals: Symphony prioritizes generating diverse molecules, while EMPP focuses on learning precise interaction patterns.

- **Prediction Horizon:** Symphony predicts the position of an atom without considering the positions of subsequent atoms. EMPP, however, uses all available positional information except for the masked atom's position. This difference again stems from their objectives: Symphony emphasizes diversity and fragment-based generation, while EMPP prioritizes accuracy by minimizing interfering factors in position prediction to capture genuine physical interactions.

- **Spatial Projection Module:** Symphony normalizes multi-channel features using the aggregation $\log(\sum_{channel} \exp(\cdot))$. While parameter-efficient, this approach may make the resulting distribution sensitive to the spherical harmonic representation, $\mathbf{f}_k$, particularly after gridding. EMPP employs a neural network (detailed in equation 14) to aggregate multi-channel features on the sphere. Ablation experiments (Appendix C.2) demonstrate the greater stability of the distribution generated by EMPP's neural network.

- **Label Definition:** Symphony uses a Dirac delta distribution as the label for spatial projection to encourage sampling of physically plausible positions. EMPP, not designed for sampling, relaxes this constraint and adopts a smoother distribution to improve generalization performance.

Table 5: Hyper-parameters for QM9 experiments without pre-train.

| Hyper-parameters | Value or description |
|---|---|
| Optimizer | AdamW |
| Learning rate scheduling | Cosine learning rate with linear warmup |
| Warmup epochs | 5 |
| Maximum learning rate | $5 \times 10^{-4}, 1.5 \times 10^{-4}$ |
| Batch size | $128, 64$ |
| Number of epochs | $300, 600$ |
| Weight decay | $5 \times 10^{-3}, 0$ |
| Dropout rate | $0.0, 0.2$ |
| Cutoff radius (Å) | 5 |
| | EMPP |
| The maximum value of degree | 2 |
| Number of Sampling | $100^2$ |
| Intermediate dimensions | $[(64,0), (64,1), (64,2)]$ |
| Output dimensions | $[(32,0), (32,1), (32,2)]$ |
| Encoding dimensions | $[(128,0), (64,1), (32,2)]$ |
| MLP on the spherical representation | $Linear(32,16) - SiLU - Linear(16,1)$ |
| Loss weights | 1 |
| Temperature | 0.1 |

## C  DETAILS OF EXPERIMENTS AND SUPPLEMENTARY EXPERIMENTS

### C.1  IMPLEMENTATION DETAILS

First, we introduce the hyperparameter configuration of EMPP when used as an auxiliary task, with the basic training configurations based on the Equiformer setup. The configurations for QM9 and MD17 are recorded in Table 5 and Table 6, respectively. Note that there are two sets of configurations for QM9, the one with a longer epoch or smaller batch size is for the four tasks of $G$, $H$, $U$ and $U_0$ tasks.

In the experiments on PCQM4MV2, we only use EPMM for pre-training. The experimental configurations are shown in Table 7. The validation set of PCQM4MV2 is used to evaluate the accuracy of position prediction. Most of our training configurations are the same as those of the Denoising method (Zaidi et al., 2023). During the fine-tuning phase on QM9, we use the training parameters from Denoising method and the model hyper-parameters in Table 7. The loss weight for EMPP is set to 1.

**Hyper-parameters of Baselines.** In Table 1 and Table 2, the results of baselines is from (Liao & Smidt, 2023). In Table 3, the results of baselines is from (Feng et al., 2023). Additionally, the result of AttrMask is from (Luo et al., 2022).

In the ablation study, we verified the relationship between the denoising method and the standard deviation. The experimental setup for this part follows Work A. The difference is that we did not introduce a PCQ-based pre-training model, and the loss weight for denoising was set to 1.

Table 6: Hyper-parameters for MD17 dataset without pre-train.

| Hyper-parameters | Value or description |
|---|---|
| Optimizer | AdamW |
| Learning rate scheduling | Cosine learning rate with linear warmup |
| Warmup epochs | 10 |
| Maximum learning rate | $5 \times 10^{-4}$ |
| Batch size | 8 |
| Number of epochs | 2000 |
| Weight decay | $1 \times 10^{-6}$ |
| Dropout rate | 0.0 |
| Weight for energy loss | 1 |
| Weight for force loss | 80 |
| Cutoff radius (Å) | 5 |
| | EMPP |
| The maximum value of degree | 3 |
| Number of Sampling | $100^2$ |
| Intermediate dimensions | $[(64, 0), (64, 1), (64, 2), (32, 3)]$ |
| Output dimensions | $[(32, 0), (32, 1), (32, 2), (32, )]$ |
| Encoding dimensions | $[(128, 0), (64, 1), (64, 2), (32, 3)]$ |
| MLP on the spherical representation | $Linear(32, 16) - SiLU - Linear(16, 1)$ |
| Loss weights | 50 |
| Temperature | 0.1 |

## C.2 SUPPLEMENTARY EXPERIMENTS

**GEOM-Drug.** GEOM-Drug (Axelrod & Gomez-Bombarelli, 2022) is a dataset containing a diverse collection of non-equilibrium molecular data, featuring a broader range of atomic types and molecular sizes. In our experiments, we use Equiformer as the backbone model. The training configuration follows the setup used in QM9 experiments, with the number of training epochs set to 300. The absolute energy of each conformation is used as the target label.

Table 7: Hyper-parameters for PCQM4MV2 dataset.

| Hyper-parameters | Value or description |
|---|---|
| Optimizer | AdamW |
| Learning rate scheduling | Cosine learning rate with linear warmup |
| Warmup steps | 10000 |
| Maximum learning rate | $5 \times 10^{-4}$ |
| Batch size | 70 |
| Number of epochs | 20 |
| Weight decay | 0.0 |
| Cutoff radius (Å) | 5 |
| | EMPP |
| The maximum value of degree | 3 |
| Number of Sampling | $100^2$ |
| Intermediate dimensions | $[(64, 0), (64, 1), (64, 2), (64, 3)]$ |
| Output dimensions | $[(32, 0), (32, 1), (32, 2), (32, 3)]$ |
| Encoding dimensions | $[(128, 0), (128, 1), (128, 2), (128, 3)]$ |
| MLP on the spherical representation | $Linear(32, 16) - SiLU - Linear(16, 1)$ |
| Temperature | 0.1 |

Table 8: Ablation experiments of multiple atom masking on QM9 tasks.

| Task | $\alpha$ | $\Delta\epsilon$ | $\varepsilon_{HOMO}$ | $\varepsilon_{LUMO}$ |
|---|---|---|---|---|
| Units | bohr$^3$ | meV | meV | meV |
| Mask 1 atom | .041 | 27 | 14 | 13 |
| Mask 2 atom | .046 | 31 | 16 | 16 |
| Mask 3 atom | .052 | 33 | 20 | 19 |

For data preparation, we randomly sample 200,000 molecules from GEOM-Drug as the training set and 10,000 molecules as the validation set. To ensure the reliability of the validation results, SMILES strings appearing in the validation set are excluded from the training set, recognizing that a single SMILES can represent multiple conformational data points.

The experimental results, summarized in Table 15, demonstrate that EMPP achieves substantial performance improvements on GEOM-Drug, reducing the energy MAE by $48\%$. From this, we draw two key conclusions: (a) EMPP is effective for more complex organic molecular systems. (b) Despite the prevalence of non-equilibrium structures in GEOM-Drug, EMPP remains highly effective as an auxiliary task.

This experiment highlights EMPP's ability to overcome the limitations of denoising methods, which are typically restricted to approximating equilibrium structures.

Table 9: Ablation experiments of spherical operations on QM9 tasks.

| Task | $\alpha$ | $\Delta\epsilon$ | $\varepsilon_{HOMO}$ | $\varepsilon_{LUMO}$ |
| --- | --- | --- | --- | --- |
| Units | bohr$^3$ | meV | meV | meV |
| Shared MLP | .041 | 27 | 14 | 13 |
| $\log(\sum_{channel} \exp(\cdot))$ | .043 | 28 | 14 | 14 |

Table 10: Ablation study on label distribution.

| Method | $\alpha$ | $\varepsilon_{HOMO}$ |
| --- | --- | --- |
| Without self-supervision | .046 | 15.4 |
| Baseline | .041 | 14.2 |
| Directly predict relative positions | .044 | 15.1 |
| Dirac delta for radius | .042 | 14.6 |
| Dirac delta for direction | .044 | 14.6 |
| Dirac delta for both | .044 | 14.8 |

**Dirac delta distribution vs. Gaussian distribution.** We have mentioned in Section 3.2.3 that whether Dirac delta distribution and Gaussian distribution are a one-to-one projection to ground truth $\vec{r}_{ik}$, they are all theoretically correct. Here, we conduct experiments to investigate their difference. In detail, we set the $\sigma$ in equation 16 to $0.0001$ to make the radius distribution similar to Dirac delta distribution (The ideal Dirac delta distribution is hard to sample). Similarly, we add the temperature coefficient to equation 17 and set it to $0.0001$ to make the spherical distribution closed to Dirac delta distribution. Additionally, We also conducted a set of experiments to directly predict relative positions $\vec{r}_{ik}$.

The results are shown in Table 10, where "Baseline" denotes using original setting. we found that the Dirac delta distribution can also achieve improvements compared to distribution in equation 16 and equation 17. However, the sharp distribution will reduce the stability of training: when we repeat the experiments, the baseline model converges to the optimal result every time, however, EMPP based on the Dirac distribution requires more than five repetitions of the experiment to find the results presented in Table 10. Finally, there is another observation from Table 10: directly predict the relative positions limits the effectiveness of EMPP.

**Transferability.** We also assessed the transferability of two distributions. We pre-trained on PCQM4MV2 and transferred to QM9. We use the TorchMD-Net model and the "baseline" employed EMPP's 1-mask strategy. "Dirac delta for both" involved changing the distributions in both pre-training and fine-tuning to Dirac delta distribution, similar to the experiments in Table 10. From Table 11, it can be seen that a relaxed distribution contributes to the transferability of EMPP.

Table 11: Ablation study on label transferability. The backbone is TorchMD-Net.

| Method | $\alpha$ | $\varepsilon_{HOMO}$ |
| --- | --- | --- |
| Without self-supervision and pre-train | .059 | 20.0 |
| Baseline | .037 | 14.3 |
| Dirac delta for both | .049 | 17.3 |

**Training time consumption.** EMPP is a self-supervised approach that generates new data indirectly, which can lead to increased training resource consumption. We have compared the computational time required for calculating the EMPP loss with other methods in Table 12. The findings indicate

that the time taken to compute EMPP is similar to that of calculating losses for property prediction or denoising tasks. Consequently, employing n-mask EMPP as an auxiliary task is expected to roughly multiply the training time by a factor of n. Based on the experimental results presented in Tables 1 and 2, we find that the 1-mask strategy is generally sufficient for most scenarios: it doubles the training time but also yields substantial performance enhancements. When computational resources and time are not constraints, we suggest opting for a higher n in the mask strategy for even better outcomes.

Table 12: Ablation study on training time consumption.

| Method | Samples per second | The cost of each iterations (ms) |
|---|---|---|
| Property prediction | 291.57 | 439 |
| Denoising | 290.25 | 441 |
| EMPP | 281.94 | 454 |

**Masking multiple atoms in a molecule.** EPMM is based on a crucial theoretical premise: when a single atom is masked, its position is well-posed in most cases, but when two atoms are masked, the position prediction becomes ill-posed. We empirically validate this assertion through experiments. We construct a variant in which multiple n atoms are masked in each molecule, and the remaining atoms' embeddings are used to predict the positions of all atoms simultaneously. From Table 8, We found that when multiple atoms were masked, the performance actually declined, and only when a single atom was masked did EMPP surpass the baseline. This experiment confirmed our concerns regarding the ill-posed nature of masking multiple atoms. Furthermore, in practical applications, we recommend the approach of repeatedly masking a single atom, which is decribed in equation 8.

Table 13: Ablation study on hyper-parameters in label distribution.

| Method | $\alpha$ | $\varepsilon_{HOMO}$ |
|---|---|---|
| Without self-supervision | .046 | 15.4 |
| Baseline ($\tau = 0.1/\sigma = 0.5$) | .041 | 14.2 |
| ($\tau = 0.1/\sigma = 0.3$) | .041 | 14.2 |
| ($\tau = 0.1/\sigma = 0.1$) | .041 | 14.3 |
| ($\tau = 0.1/\sigma = 0.05$) | .042 | 14.2 |
| ($\tau = 0.1/\sigma = 0.01$) | .042 | 14.3 |
| ($\tau = 0.5/\sigma = 0.5$) | .041 | 14.2 |
| ($\tau = 0.3/\sigma = 0.5$) | .041 | 14.1 |
| ($\tau = 0.05/\sigma = 0.5$) | .042 | 14.3 |
| ($\tau = 0.01/\sigma = 0.5$) | .042 | 14.4 |

**Operations on Sphere.** In Section 3.2.2, we mentioned that operations on the sphere do not affect equivariance. In order to map the information of all channels to 1D, we employed a shared fully connected layer. Previous methods have used aggregation $\log(\sum_{channel} \exp(\cdot))$ for mapping to 1D. It can be observed from the Table 9 that the fully connected layer introduces more parameters, but it also enhances performance. Moreover, parameter-free methods may encounter numerical overflow issues during the training process.

Table 14: The difference between prediction based on neighbor structure and global molecular graph.

| Task | $\alpha$ | $\varepsilon_{HOMO}$ | $\varepsilon_{LUMO}$ |
|---|---|---|---|
| Units | bohr$^3$ | meV | meV |
| EMPP (Neighbor) | .041 | 14 | 13 |
| EMPP (Global) | .045 | 118 | 16 |
| EMPP (Global + Dist Weights) | .044 | 15 | 14 |

**The impact of hyper-parameters in label distribution.** $\tau$ in equation 14 and $\sigma$ in equation 15 are two important hyperparameters, and we conducted ablation studies to evaluate them. This ablation

study was not based on pre-training and used Equiformer to train on QM9 with EMPP as an auxiliary task. From Table 13, we found that when we changed these two hyperparameters, there was no significant change in performance, which reflects that EMPP is not sensitive to the distribution of labels.

Table 15: Results on GEOM-Drug property prediction without pre-trainging.

| Method | Energy MAE (kcal/mol) |
|---|---|
| Equiformer | 0.07517 |
| Equiformer+EMPP | 0.03912 |

**Prediction with all the unmasked atoms.** In EMPP, we use the embeddings of neighboring nodes to predict the position of the masked atom. However, in reality, all atoms interact with the masked atom, even if the interactions are minimal. Here, we have designed two variants of EMPP. The first variant uses all atoms to predict the position ("Global"), and the second variant uses all atoms to predict and adds distance-related weights ("Global + Dist Weights"). In other words, the second method weights the loss generated by the first variant, with greater weight given to atoms that are closer to the masked atom. The weighting function is $w_k = \max(0, \frac{|\vec{r}_{ik}| - C}{C})$, where $C$ is a hyper-parameter set to 8 Å. From Table 14, we observe that utilizing all unmasked atoms to predict positions diminishes the model's generalization capability. We believe this may be due to inaccurate long-range interactions affecting the model's learning. Although incorporating distance weights improves the global prediction effect, it still fails to match the performance of predicting based on neighboring atoms.

**The impact of molecules of different sizes on EMPP.** EMPP improves its generalization performance by masking atomic positions and then restoring them. Intuitively, as the number of atoms in a molecule increases, more atoms can be masked, allowing EMPP to generate more diverse data. To evaluate the impact of EMPP on molecules with different sizes, we conducted experiments by categorizing the QM9 training data into three groups based on the number of atoms: (0-16), (17-19), and (20+). These categories contain roughly equal amounts of data. In each experiment, we computed the EMPP loss using only molecules from one of these categories. As shown in Table 16, the results demonstrate that EMPP consistently improves performance across different molecular sizes, with larger molecules experiencing more significant performance gains.

Table 16: Ablation study on the impact of molecules of different sizes.

| Method | $\alpha$ | $\varepsilon_{HOMO}$ |
|---|---|---|
| Baseline | .041 | 14.2 |
| EMPP on (0-16) | .044 | 14.9 |
| EMPP on (17-19) | .043 | 14.4 |
| EMPP on (20+) | .043 | 14.5 |

