# OpenReview forum: "Equivariant Masked Position Prediction for Efficient Molecular Representation"
_ICLR.cc/2025/Conference — ICLR 2025 Poster_

### Official Review · Reviewer_RrYB · 2024-10-26

**Soundness:** 2
**Presentation:** 3
**Contribution:** 3
**Rating:** 6
**Confidence:** 4

**Summary:**

This paper proposed EMPP, a new approach for pre-training GNNs for 3D molecules with equivariant mask positional prediction. Instead of directly denoising the position from the approximation of Gaussian mixtures, the authors proposed to model the distribution over positions with equivariant GNNs. Downstream tasks on QM9 and MD17 demonstrated improvements in the performance after such a pre-training.

**Strengths:**

- The idea of predicting distributions over positions instead of directly predicting the atom positions is novel. Previous work, as was demonstrated in the paper, predominantly relied on the direct prediction of the mask atoms.
- Experimental results on QM9 and MD17 demonstrated the improvement of pre-training compared to the baseline models, in both end-to-end training and pre-training settings.
- An anonymous code link is provided for better reproducibility.

**Weaknesses:**

1. The delivery of this paper is poor, with noticeable inconsistencies regarding the proposed method, irrelevant materials and experiments.
    - **Irrelevant related work**. The proposed method has nothing to do with language models (not even the language modeling for chemical sequences like SMILES). However, the authors started with LLM, which was misleading and confusing.
    - **Irrelevant experiments**. The ablation study in Section 5.4 demonstrated the results for denoising pre-training models, but not for the proposed method in the paper. In contrast, the ablation study in Appendix C was designed for the proposed model. It is unclear why the authors put the irrelevant ablation studies in the main text.
    - **Objective in pre-training**. In Figure 1 and Section 3.2.1, the authors seemed to indicate their proposed approach is a force-prediction network for pre-training. However, Figure 2 and Section 3.2.2 indicated that the displacement vector $r_{ij}$ was instead regressed as the learning target.
    - **Objective in finetuning**. In Section 3.3, the energy and force prediction seemed to be the auxiliary input for pre-training. However, in the QM9/MD17 setting, they are the prediction targets. It is unclear how the model was trained end-to-end without the proposed pre-training objective.

2. **The claim of Gaussian mixture issues is questionable**. The authors claimed that the proposed approach can "effectively avoid
the approximation of Gaussian mixture" (Section 3.2.1). However, the loss functions in Eq.14 also explicitly rely on the Gaussian mixture assumption to derive the KL loss. The direction loss in Eq.16 assumes an alternative form based on the softmax function. The only difference is the Gaussian is defined for the radii and angles instead of Euclidean coordinates.

3. **The innovation in force prediction is lacking**. It is unclear about the major motivation behind the proposed method of "force prediction". Previous work (e.g., [1], also cited in the paper, [2], [3]) has clearly indicated the relation between the potential energy surface and the force field prediction with equivariant GNNs. In this sense, the proposed approach is exactly the same as other denoising pre-training models. Furthermore, the indicated issues with the Gaussian mixture of variable variance landscape can be effectively addressed with iterative sampling methods (i.e., diffusion models like [4]) and annealed Langevin dynamics. Alternatively, [2] introduced invariant noise-scale prediction which can handle different noise scales.

4. **The local equivariant prediction of the position distribution is not consistent as a pre-training objective** (unless the Gaussian variance is 0). This can be easily demonstrated by the fact that the radii and direction angles are constructed with respect to different neighboring atoms such that the expanded ground truth distribution roughly follows the shape of a small region of a spherical shell. For neighboring atoms at different positions, the ground truth is inconsistent unless the variance is 0 (i.e., a Dirac distribution).


[1] Zaidi, Sheheryar, et al. "Pre-training via denoising for molecular property prediction." arXiv preprint arXiv:2206.00133 (2022).

[2] Jiao, Rui, et al. "Energy-motivated equivariant pretraining for 3d molecular graphs." Proceedings of the AAAI Conference on Artificial Intelligence. Vol. 37. No. 7. 2023.

[3] Jiao, Rui, et al. "Equivariant Pretrained Transformer for Unified Geometric Learning on Multi-Domain 3D Molecules." arXiv preprint arXiv:2402.12714 (2024).

[4] Hoogeboom, Emiel, et al. "Equivariant diffusion for molecule generation in 3d." International conference on machine learning. PMLR, 2022.

**Questions:**

1. What is the training objective in the end-to-end training for the QM9/MD17 dataset? See Weakness 1.
2. Why can the proposed framework address the Gaussian mixture issue? In your pre-training objectives, Gaussian assumptions (or other softmax-based assumptions) were also explicitly assumed to make the ground truth distribution target. See Weakness 2.
3. What is the major innovation of the proposed framework as a "force prediction" model? Previous work has clearly indicated the close relationship between force field prediction and the denoising process of atom positions. Equivariant GNNs were also used in these works. In this sense, the approach proposed in this work is essentially the same as existing denoising frameworks. See Weakness 3.
4. How do you address the issue of inconsistent pre-training objectives if Gaussians are assumed independently for radii and angles? See Weakness 4.
5. What is the impact of $\tau$ in Eq.13? What is the impact of $\sigma$ in Eq.14? According to the ablation study of the denoising baselines, these values that control the "sharpness" of the distributions have a large impact on the final performance. However, they were not fully tested in the paper with the proposed approach.

---

> ### Author Response · Authors · 2024-11-18
> **Responses to Reviewer (Main concerns, Weakness 1)**
>
> Thank you very much for your review. **We will address all your concerns, and to save your time, we will bold some key points.**
>
> ## Response to main concerns:
>
> **(1) The Gaussian distribution in Equation 14 is used to transform scalar distances $|\vec{r}\_{ij}|$ into a set of continuous and smooth features. This transformation enhances the model's expressive power and makes it easier to learn complex geometric relationships. This approach is known as Gaussian RBF, which is commonly used in equivariant GNNs, such as TorchMD-Net [1], TFN [2], SCN [3]. The directional distribution in Equation 16 serves a similar purpose. It is important to emphasize that in EMPP, the true labels are the well-defined relative positions $\vec{r}\_{ij}$. As long as the defined distribution can uniquely represent $\vec{r}\_{ij}$, the model can learn the correct information. Moreover, we have included additional experiments in Appendix C.2 (Table 10) to show that EMPP with a Dirac delta distribution also work well. However, the smooth distributions we employed can enhance training stability and yield superior results. Further details are provided in the subsequent responses. Additionally, we have revised the loss section to clarify potential misunderstandings. Thank you for your suggestion.**
>
> [1] TorchMD: A deep learning framework for molecular simulations. Stefan Doerr, et al.
>
> [2] Tensor field networks: Rotation- and translation-equivariant neural networks for 3D point clouds. Nathaniel Thomas, et al.
>
> [3] Spherical Channels for Modeling Atomic Interactions. C. Lawrence Zitnick, et al.
>
> **(2).Our method is fundamentally different from denoising approaches. Denoising methods approximate an unknown potential energy surface (PES) by using Gaussian mixtures and compute its derivative (around noisy points) to create a pseudo-label for the force prediction. In contrast, EMPP does not approximate unknown physical variables. It bypasses the PES and its derivatives, instead directly modeling the relationship between atomic interactions and positions. Since the force field is a function of position that reflects overall interactions, position prediction in EMPP can indirectly capture the key features of the force field.**
>
> ## Responses to Weakness 1:
>
> We have submitted a revised paper to address these concerns, which you can check. Additionally, we would like to make some explains:
>
> (1) The role of the language model is to highlight the importance of self-supervision and data augmentation. We have modified this part to enable the readers to understand clearly.
>
> (2) The ablation study in Section 5.4 includes two experiments (Table 4 and Figure 3). **Table 4 is for the proposed method and Figure 3 reflects the fundamental shortcomings of denoising which is very important for understand our motivation.** Due to space limitations, we move all other less critical ablation to the appendix.
>
> (3) We clarify that EMPP is not to predict forces. **The purpose of EMPP is to enable models to predict a physically plausible position using a set of neighbors, which essentially requires the model to implicitly model the correct atomic interactions (such as force field).** Besides, there exists a difference: **denoising methods approximate the PES and computing derivatives to learn forces, while EMPP models the functions of atomic interactions and atomic absolute position.**
>
> (4) During fine-tuning, EMPP serves as an auxiliary task, meaning each training molecule passes through the network twice: **first to calculate the regular energy or forces loss, and second to calculate the EMPP loss, with both contributing to gradient descent. The denoising methods are also as a similar auxiliary task during fine-tuning [4,5].** The difference is that EMPP can model the relationship between labels and molecular structures (positions), thus promoting generalization of backbone to certain labels, whereas denoising as an auxiliary task does not consider labels. **Our experiments (Table 1, Table 2) also demonstrate that even without pre-training (introducing additional data) EMPP can enhance model generalization, whereas denoising methods lose effectiveness without pre-training, as shown in the table below.**
>
> (* denotes the results we that we have reproduced in the same environment.)
> ｜ Index | Method      | $\alpha$ | $\Delta \epsilon$ | $\varepsilon_{HOMO}$ | $\varepsilon_{LUMO}$ |
> | ----------- | ----------- | ----------- | ----------- | ----------- | ----------- |
> | (i) | *TorchMD-Net (Without pretrain and denoising) | .0593 | 36.4 | 22.3 | 21.4 |
> | (ii) | DP-TorchMD-Net （Pretrain）| .0517 | 31.8 | 17.7 | 14.3 |
> | (iii) | *DP-TorchMD-Net （Without pretrain）  | .0579 | 35.6 | 20.5 | 19.9 |
> | (iv) | *EMPP + TorchMD-Net （Without pretrain）  | .0546 | 33.7 | 18.6 | 16.0 |
>
> [4] Pre-training via denoising for molecular property prediction. Zaidi S, et al.
>
> [5] Fractional denoising for 3d molecular pre-training. Feng S, et al.

---

> ### Author Response · Authors · 2024-11-18
> **Responses to Reviewer (Weaknesses 2-4, Question 1-5)**
>
> ## Responses to Weakness 2:
> The Gaussian distribution in our mehtod is different from it in denoising. Denoising methods use it to approximate the local minima of unknown PES. We use it to represent the well-defined relative positions $|\vec{r}\_{ij}|$, like the Gaussian RBF in previous molecular models (See Responses to main concerns). The spherical representation in Equation 16 is similar, They brings a significant benefit: **facilitating learning of neural networks. For example, The computer represents distributions using sampling on the radius or sphere, and the peak of the strict Dirac distribution might be missed by finite sampling. When we use smooth representation like Gaussian, computer sampling can better fit mathematical distributions, making the loss calculation accurate.** The experiments in Table 10 show that training with smooth functions is more stable and easier to achieve better results.
>
> ## Responses to Weakness 3:
> The explanation of "force prediction" can be found in responses (3) to Weakness 1. Besides, EMPP is fundamentally different from denoising methods (See Responses to main concerns).
>
> The implementations of EMPP and denoising are also significantly different: **denoising methods use noise sampling to predict noise, whereas EMPP employs position masking and predicts positions based on unmasked atoms.** In denoising, equivariant GNNs are primarily used to enforce the equivariance of the predicted noise. In contrast, **we use equivariant models for an additional purpose: equivariant models are capable of describing complex interactions (Theorem 2 in [6]), particularly when the degree of interaction is high, which enables EMPP to accurately model the relationship between interactions and positions.**
>
> [6] On the Universality of Rotation Equivariant Point Cloud Networks. Nadav Dym, Haggai Maron
>
> As for your point on diffusion, it is similar to denoising methods, but not the same. **The similarity arises because predicting the forces from the Boltzmann distribution is precisely a score matching process [4].** However, the purpose of EDM (which you referenced as [4] in reviews) is to add artificially defined noise to the input and then recover the molecular structure from the noise, with all distributions in diffusion being artificially defined and known. **The goal of denoising is not to generate molecules, but to produce plausible data to enhance generalization. However, the real physical spaces (such as PES) are unknown and denoising methods use Gaussian to approximate. Therefore, the challenge lies not in solving score matching, but in how to approximate the real physical distribution.** Similarly, the citation [2] in your reviews predicts the $\sigma$ of Gaussian; however, the label of $\sigma$ used during training is still an approximation, not the curvity of true physical distribution.
>
> ## Reponses to Weakness 4:
> Please refer to Responses to main concerns and Weakness 2 to understand the role of Gaussian distribution we use. Additionally, you provided a great example. **Please note that the purpose of the predicted distribution is to describe the ground truth $\vec{r}\_{ij}$, not generate atoms or fit the unknown distribution. Thus, we do not need to align the distribution of each neighbor; as long as it can represent $\vec{r}\_{ij}$, that is enough.** Equation 14 and 16 are both unique mappings of $\vec{r}\_{ij}$.
>
> ## Responses to Question 1:
> Please refer to our responses (4) to Weakness 1.
>
> **Thank you. We have provided a clearer explanation of fine-tuning in the revised version, indicated in red, please review.**
>
> ## Responses to Question 2:
> Please refer to our responses (4) to Weakness 2.
>
> **Thank you. We have provided a clearer explanation of the Gaussian assumption in the revised version, indicated in red, please review.**
>
> ## Responses to Question 3:
> Please refer to our responses to Weakness 3.
>
> ## Responses to Question 4:
>
> Please refer to our responses to Weakness 4. As we mentioned, we do not need to align the distributions of neighbors. **These distributions are not intended to generate or fit real distributions. They are projections of $\vec{r}\_{ij}$, which is real in molecule and consistent for all neighbors.**
>
> ## Responses to Question 5:
>
> Thank you for your question. **We have added experiments regarding $\tau$ in the Appendix C.2 (Table 13). Briefly, EMPP is not sensitive to $\tau / \sigma$. we generally use the common setting of $0.1/0.5$, and our additional experiments also show that the common setting is suitable for different tasks, and for different values, results are similar.** As previously mentioned, the role of these hyper-parameters are not same to $\sigma$ in denoising (which is used to approximate unknown curvity). Thus, They can be flexibly determined.
>
> Thank you for your review. Our revision will not affect any core ideas and implementation, but help make the paper clearer. If you have any other concerns, please do not hesitate to let us know.

---

> > ### Comment · Reviewer_RrYB · 2024-11-22
> > **Decision to raise my scores**
> >
> > I thank the authors for their detailed responses regarding my existing concerns. I also checked the updated manuscript and was glad to find out that the authors have followed my suggestion to improve the presentation of the paper. Furthermore, as the distinction from common diffusion-based (denoising) models has been made clear in the revised manuscript with additional experimental results, I now believe this work can be moderately interesting to the machine learning community. Therefore, I raise my score from 3 to 6.

---

### Official Review · Reviewer_DiCi · 2024-10-27

**Soundness:** 2
**Presentation:** 3
**Contribution:** 2
**Rating:** 6
**Confidence:** 4

**Summary:**

This paper introduces a self-supervised learning approach called Equivariant Masked Position Prediction (EMPP) for molecular graph neural networks. The main idea behind is to mask the 3D position of atoms while keeping their other attributes, then predict these positions using the surrounding molecular structure. The authors intend to address the limitations of other self-supervised approaches. Results show comparable/superior performance to other state-of-the-art approaches for quantum-mechanical property prediction.

**Strengths:**

The core idea of predicting 3D atomic positions while keeping atomic attributes seems a reasonable evolution of existing molecular self-supervised approaches, taking into account the claimed limitations of previous approaches. The paper's strongest aspect is its clear presentation. The problem motivation and methodology are well articulated. The experimental validation, while having some gaps, provides adequate evidence on standard benchmarks. These strengths suggest incremental results.

**Weaknesses:**

The performance improvements on MD17 are relatively modest compared to the baseline Equiformer. This raises questions about whether the added complexity of EMPP is justified by the gains. Furthermore, for non-pretaining comparisons, there are models that substantially outperform the proposed method, based also on higher-order tensors.

Also, the computational overhead of EMPP isn't adequately addressed. The method requires calculating distributions over a spherical grid (100² points according to Table 7) for each masked atom, which could be computationally expensive. An analysis of training time and memory requirements compared to simpler approaches is missing.

I feel that altering the Equivariant Transformer architecture while testing the validity of the method simultaneously makes it difficult to distill where the improvements come from.

**Questions:**

1) What is the computational overhead of calculating distributions over the 100² spherical grid points compared to simpler approaches? Perhaps providing training time and memory requirements compared to baseline methods could also help.

2) How sensitive are the results to the sampling rate on the sphere?

3) Providing baseline results using the modified ET from TorchMD-Net could help distill how significant is the improvement due to the inclusion of the masking task. (By the way, the citation of the ET architecture is wrong).

---

> ### Author Response · Authors · 2024-11-18
> **Responses to Reviewer**
>
> Thank you very much for your review. Your concerns and questions are very meaningful. **In the revised version, we have added experiments on time cost (Appendix C.2, Table 12).** You can also directly look at the response below to save time.
>
> ## Responses to Questions 1:
> We have established three tasks: **1. Only predicting the alpha property of QM9 using Equiformer, 2. Only predicting positions using Equiformer (Ours), 3. Only predicting noise using Equiformer (Denoising).** They correspond to three models with different output heads. Each experiment was conducted on a single Nvidia A100, and the table below shows that the time consumptions of three models are similar (EMPP is slightly bigger). **We argue there are two main reasons why the time consumption introduced by EMPP's output head is small: First, the output head of EMPP only considers the neighboring nodes of the masked atoms, and the rest of the nodes do not need computation. Second, before entering the grid module, the number of embedding channels has already been compressed to a small number 32 (as we mentioned in Table 5 and 6).**
>
> | Index | Method | Samples per second | The cost of each iterations (ms) |
> | ----------- | ----------- | ----------- | ----------- |
> | (i) | Property prediction | 291.57 | 439 |
> | (ii) | Denoising | 290.25 | 441 |
> | (iii) | EMPP  | 281.94 | 454 |
>
> Additionally, you mentioned about the small improvement on MD17. We consider that is because: the training set of MD17 is a small (950 molecules) and easy-to-learn dataset. Thus, the MAE of Equiformer has reached saturation (less than 5 meV in most tasks), and the augmentation of EMPP for 950 data may be limited. **The more complex QM9 dataset can better demonstrate the advantages of EMPP.**
>
> I apologize for any confusion. In your mention of weaknesses, you state, 'for non-pretraining comparisons, there are models that substantially outperform the proposed method, based also on higher-order tensors.' If you're referring to the pre-training comparison in Table 3, we used TorchMD-Net for the comparison with denoising methods (since they used it as the backbone). **In fact, EMPP can be applied to any equivariant GNNs.**
>
> ## Responses to Questions 2:
> We have published some of the sampling results in the Table 4. In our task, we want to represent fine-grained spherical distribution with dense sampling, and the grid operation does not actually take up a significant amount of time, so we use a larger sampling rate to ensure the results. In fact, **when the sampling rate exceeds a certain value, the performance will stabilize**: we tested the sampling rate of 80/90 during the rebuttal period, the MAE of $\\alpha$ and $\varepsilon_{HOMO}$ are 0.041/0.041 and 14.4/14.3 (the results of default setting 100 are 0.041 / 14.2).
>
> ## Responses to Questions 3:
>
> Good question! We conducted experiments to verify whether the improvement comes from our TorchMD variant. As shown in the table below, **the modified TorchMD achieved very similar results to the original.** This is mainly because that we only projected the node embeddings onto higher orders using spherical harmonics without changing the core operations of ET. You can also see from our Table 1 and Table 2 that EMPP introduced no changes on the Equiformer, yet it still achieved improvements. Additionally, we modified TorchMD because higher-order equivariance helps to describe a finer-grained spherical function (after grid module), which thus aids in the learning of EMPP.
>
> (* denotes the results we that we have reproduced in the same environment.)
>
> | Index | Method      | HOMO | LUMO |
> | ----------- | ----------- | ----------- | ----------- |
> | (i) | *TorchMD-Net （Without pretrain）  | 22.3 | 21.4 |
> | (ii) | *TorchMD-Net + Higher-degree （Without pretrain）   | 22.0 | 21.1 |
> | (iii) | *TorchMD-Net + Higher-degree + EMPP （Without pretrain）  | 18.6 | 16.0 |
>
> Finally, we greatly appreciate the pointing out of our citation error, which has been corrected. All of your questions are of significant importance in refining our paper.

---

> > ### Comment · Reviewer_DiCi · 2024-11-21
> > **Reply**
> >
> > Thank you for your response.
> >
> > I am satisfied with how the authors have addressed my concerns. The timing benchmarks seem to demonstrate that EMPP's computational overhead is minimal, with well-explained reasons for this efficiency (neighbor-only computations and channel compression). The ablation studies effectively isolate the source of improvements, showing that the modified Equivariant Transformer architecture (I insist on the reference; TorchMD is a MD framework, not a model framework), achieves similar baseline results to the original. The sampling rate analysis provides good evidence for the stability of your method.
> >
> > Overall, these clarifications and additional experiments have strengthened my confidence in the paper's contributions. I raise my score to 6.

---

### Official Review · Reviewer_Koif · 2024-10-31

**Soundness:** 3
**Presentation:** 4
**Contribution:** 4
**Rating:** 6
**Confidence:** 2

**Summary:**

This paper presents a novel self-supervised framework for molecular representation learning, named Equivariant Masked Position Prediction (EMPP). EMPP pretrains the model by directly predicting the masked positions of atoms and addresses limitations in existing self-supervised methods (attribute masking and denoising). Experimental results show EMPP achieves competitive performance. EMPP can also enhance supervised learning when used as an auxiliary task.

**Strengths:**

- The paper is well-organized. The method has clear explanations.
- The proposed EMPP approach is simple yet effective, achieving state-of-the-art on benchmark datasets.

**Weaknesses:**

In Section 5.2, the comparison does not seem entirely fair. EMPP serves as an auxiliary task and relies on the Equiformer backbone, yet it is compared with other architectures that lack this component.

**Questions:**

In Equation 8, does the averaging over multiple masked atoms effectively act as training for a longer time? How does it impact the efficiency of the model? Could you please provide empirical results comparing training times and computational efficiency between single and multiple masked atoms? Additionally, discussing potential trade-offs between performance gains and computational costs would be appreciated.

---

> ### Author Response · Authors · 2024-11-18
> **Responses to Reviewer**
>
> Thank you for your review and recognition of our paper. We will address your concerns.
> ## Responses to Weekness:
> **In Table 1 and 2, we believe that auxiliary tasks without pre-training provide more accurate reflection of the method's performance.** However, while previous denoising methods have been used as auxiliary tasks during fine-tuning, their reported results all rely on pre-training and torchMD-Net (we use Equiformer in Table 1 and 2). To provide you a more fair comparison, We conducted experiments without pre-training based on TorchMD-Net (using the open-source implementation of the denoising method [1]). The results are shown in below. **These experiments revealed that the performance improvement of denoising methods is limited in the absence of pre-training. In contrast, EMPP demonstrated a stronger improvement compared to denoising methods.**
>
> (* denotes the results we that we have reproduced in the same environment.)
>
> | Index | Method      | $\alpha$ | $\Delta \epsilon$ | $\varepsilon_{HOMO}$ | $\varepsilon_{LUMO}$ |
> | ----------- | ----------- | ----------- | ----------- | ----------- | ----------- |
> | (i) | *TorchMD-Net (Without pretrain and denoising) | .0593 | 36.4 | 22.3 | 21.4 |
> | (ii) | *DP-TorchMD-Net （Without pretrain）  | .0579 | 35.6 | 20.5 | 19.9 |
> | (iii) | *EMPP + TorchMD-Net （Without pretrain）  | .0546 | 33.7 | 18.6 | 16.0 |
>
> [1] Pre-training via denoising for molecular property prediction. Zaidi S, et al.
>
> ## Responses to Question:
> Yes, masking multiple atoms leads to increased training time. **In our response to Questions 1 of reviewer DiCi, we evaluate the time cost of EMPP. In short, when using n-mask EMPP as auxiliary task, the training time will be expand by n times (as shown in the below tabel).** We are also glad to discuss the trade-off: **in most cases, the 1-Mask method is sufficient, as it doubles the training time but also significantly improves performance.** More masks can further enhance performance, but currently, their improvements are not significant, and the training time is expanded several times. We consider that when resources and time are abundant, multi-mask methods can be considered. Besides, EMPP does not affect the efficiency of model inference.
>
> **Our ablation study and discussion on trade-offs between performance and computational costs has been added in Appendix C.2 (Table 12).**
>
> | Index | Method | Samples per second | The cost of each iterations (ms) |
> | ----------- | ----------- | ----------- | ----------- |
> | (i) | Property prediction only | 291.57 | 439 |
> | (ii) | Denoising only | 290.25 | 441 |
> | (iii) | EMPP only | 281.94 | 454 |
> | (iv) | EMPP (1-Mask) +  Property prediction | 146.62 | 873 |
> | (v) | EMPP (3-Mask) +  Property prediction  | 71.71 | 1785 |

---

> > ### Comment · Reviewer_Koif · 2024-11-26
> >
> > Thank you for your response. I keep my original rating of 6.

---

### Official Review · Reviewer_cKne · 2024-11-02

**Soundness:** 4
**Presentation:** 4
**Contribution:** 4
**Rating:** 8
**Confidence:** 3

**Summary:**

This paper introduces the Equivariant Masked Position Prediction (EMPP) model, a self-supervised learning approach for molecular representation aimed at improving molecular property prediction in graph neural networks (GNNs). EMPP differs from conventional methods by predicting the relative position of masked atoms based on surrounding atomic structures rather than masking node attributes. This avoids the Gaussian mixture approximation used in denoising methods, potentially enhancing accuracy. EMPP is presented as a versatile technique that can function both as a pre-training task on unlabeled data and as an auxiliary task for labeled property prediction, achieving strong results across molecular benchmarks.

**Strengths:**

- **Originality**: EMPP’s approach of modeling relative atomic positions marks a novel shift from traditional self-supervised molecular methods, which typically rely on attribute masking or denoising with Gaussian mixtures. By avoiding Gaussian approximations, EMPP proposes a fresh method that could more precisely capture quantum mechanical properties without prior assumptions on potential energy surface shape.
- **Quality**: The methodology and experiment sections are comprehensive, offering detailed descriptions of the model architecture and training procedures. EMPP’s application across QM9 and MD17 benchmarks highlights its effectiveness and consistency in molecular property prediction.
- **Clarity**: The paper is clearly structured, and the motivations for EMPP are presented well. The provided comparisons, particularly with denoising methods, effectively illustrate the advantages of EMPP.
- **Significance**: EMPP’s adaptability for both pre-training and auxiliary tasks in GNNs is particularly valuable, as it could streamline the use of self-supervised learning in various molecular and materials science applications.

**Weaknesses:**

- **Unclear Evaluation in Pre-training**: EMPP is proposed for both pre-training and auxiliary loss tasks, yet performance declines are observed in some properties (e.g., HOMO, LUMO, G, H) when used as a pre-training task (see Table 1 vs. Table 3). This suggests a possible limitation in generalizability when used for pre-training. I guess this is due to dataset constraints, as QM9 may not properly diverse molecular dataset to evaluate the most recent methods.

- **Too simple benchmark** : The molecules in the benchmark test sets, particularly QM9 and MD17, are relatively small or simple compounds, which may not sufficiently assess EMPP’s performance on more larger and realistic molecular structures. This makes it challenging to evaluate EMPP's generalization capabilities for applications requiring more intricate molecular structures.

To address this, the reviewer's suggestion:

**Conduct experiments on the Geom-Drugs dataset**: the dataset includes larger and more complex molecular structures than QM9. This dataset can provide a more realistic test of EMPP's ability to generalize to more diverse molecular systems. Especially, please perform evaluations with and without pre-training to compare how EMPP's pre-training performance translates to more complex molecules in this dataset. This will help determine if the performance drop observed in QM9 pre-training persists or changes with different datasets.

**Analyze Performance Scaling with Molecular Complexity and Size**:
Extend experiments across molecules of varying complexity and size to observe how EMPP scales. This may involve assessing EMPP’s accuracy, efficiency, and convergence with increasing molecular complexity.
Present scaling analysis results, ideally quantifying any trends where EMPP’s accuracy vary with molecular size. This will provide insights into EMPP's robustness and applicability to more complex molecular systems.

**Questions:**

- **Physical Assumptions in Modeled Distributions**: It is interesting to understand the foundational physical or chemical assumptions behind EMPP’s use of relative position modeling for capturing atomic interactions. Denoising methods typically rely on a Boltzmann distribution near local minima, with harmonic potential assumptions. By contrast, EMPP’s approach seems to focus on relative distances and directional distributions between masked and neighboring atoms. This reviewer would like clarification on the theoretical basis for this choice. Specifically, do the authors view this approach as a distinct modeling assumption, or is it comparable to the physical basis provided by Boltzmann distributions in denoising methods?

- **Comparison with Denoising Methods**: Given that both EMPP and denoising methods model unknown data distributions, EMPP’s key distinction is its focus on relative, rather than absolute, atomic positions. The authors claim that one of the reasons for the performance improvement over the existing method is that they avoided the erroneous gaussian approximation (Figure 1-(d)). Then, can we expect the same level of performance improvement even if we do not adopt the method of modeling the distance and angular parts separately in EMPP? Is it possible to adopt a method that directly estimates the positional difference vector? Is it possible that the good performance of the model is due to modeling the distance and angular part separately? Please conduct ablation experiments on this.

---

> ### Author Response · Authors · 2024-11-18
> **Responses to Reviewer**
>
> Thank you for your review and recognition of our paper. We will address your concerns.
>
> ## Responses to Weakness "Unclear Evaluation in Pre-training":
> We acknowledge your guess that the poor ddiversity of QM9 limit methods. Additionally, **there is an another reason: Table 1 and Table 3 utilize different backbones (Equiformer and TorchMD-Net).** These models inherently introduce performance differences. The Equiformer in Table 1 is a more advanced model and performs better. In Table 3, **we chose TorchMD because previous Denoising methods were based on TorchMD.** To facilitate a fair comparison, we used TorchMD-Net.
>
> ## Responses to Suggestion and Weakness "Too simple benchmark":
> Great suggestion! **we will conduct the experiments on "Geom-Drugs". If it is completed by the rebuttal deadline, we will add it in the paper. Otherwise, we will make it publicly available on GitHub (Sorry, we need time and resource to train).** Now, we can first discuss other concerns.
>
> Note that EMPP is a self-supervised method, and **its purpose is to produce reasonable data within a given molecular system.** From Table 1, we observe: **the QM9 molecular system is relatively simple and lacks of diversity, and the performance (MAE) of Equiformer is approaching saturation, yet EMPP can still produce a significant performance improvement.** We believe this is sufficient to demonstrate the effort of EMPP. **If we increase the complexity and diversity of the molecular system as you mentioned, EMPP should intuitively provide even greater improvements, since it can produce more kind of reasonable data.**
>
> ## Responses to Questions "Physical Assumptions in Modeled Distributions":
> **We view our approach as a distinct modeling assumption, different from Denoising.**
>
> **Denoising methods use Gaussian mixtures to approximate an unknown (Boltzmann) distribution.** They then perform noise sampling on this distribution to learn the force field (the derivative of the PES). **EMPP does not involve any approximation to unknown physics distribution.** It models directly at the level of the force field without computing derivatives. **The purpose of EMPP is to enable models to predict a physically plausible position using a set of neighbors, which essentially requires the model to implicitly model the correct atomic interactions (such as force field).**
>
> ## Responses to Questions "Comparison with Denoising Methods":
> Note that the distribution we used is not to fit an unknown distribution, but to transform scalar distances $|\vec{r}\_{ij}|$ into a set of continuous and smooth features. This transformation known as Gaussian RBF can enhance the model's expressive power and makes it easier to learn complex geometric relationships. **We have conducted corresponding ablation studies and added the results to Appendix C.2 (Table 10). In short, we used various functions to represent relative positions, all of which achieved similar improvements. However, when we directly predict the relative positions, the improvement of EMPP was small.** We believe this is because the transformation $\vec{r}\_{ij}$ to a smooth function facilitates learning the information of relative positions. Moreover, this is also a common practice, in many molecular GNNs [1,2], models do not directly map vectors to length and direction, but use a smooth Gaussian RBF to map their length before feedding to the model. More details on why we use such distributions can be referred to our responses to weekness 2 of reviewer RrYB.
>
> [1] TorchMD: A deep learning framework for molecular simulations. Stefan Doerr, et al.
>
> [2] Tensor field networks: Rotation- and translation-equivariant neural networks for 3D point clouds. Nathaniel Thomas, et al.

---

> ### Author Response · Authors · 2024-11-22
> **Adding experiments on GEOM-Drug and Analyzing Performance Scaling**
>
> **We conducted experiments on GEOM-Drug during the rebuttal period, and the introduction and analysis of this part have been incorporated into the revised version (appendix C.2).** We use Equiformer as the backbone model. The training configuration follows the configuration on QM9 experiments, with the number of training epochs set to 150. We use the absolute energy of each conformation as the label. Additionally, we randomly sample 200,000 molecules from GEOM-Drug as the training set and 10,000 as the validation set (Apologies for the limited resources and time; we are unable to use the full dataset. However, 200,000 data points should still be sufficient to demonstrate the advantages of EMPP.). To ensure the validity of the validation results, SMILES that appear in the validation set do not appear in the training set (a single SMILES includes multiple conformational data). **The below table show that EMPP can achieve significant performance improvements on GEOM-Drug. The energy MAE can be decreased by $32$\%. We draw two conclusions: 1. EMPP is applicable to more complex organic molecular systems; 2. Most molecules in GEOM-Drug are non-equilibrium. EMPP can still achieve an significant improvement when used as an auxiliary task.** This experiment shows that EMPP overcomes the limitation of denoising methods that can only approximate local minima (equilibrium structure)."
>
> | Index | Method      | Energy MAE (kcal/mol)|
> | ----------- | ----------- | ----------- |
> | (i) | Equiformer | 0.1601 |
> | (ii) | Equiformer + EMPP  | 0.1094 |
>
>
> [Updated in 11.25, We trained the Drug dataset for 300 epochs to achieve a stable performance. The results show that EMPP can produce a more significant improvement **(48%)**.]
>
> | Index | Method      | Energy MAE (kcal/mol)|
> | ----------- | ----------- | ----------- |
> | (i) | Equiformer (300 epochs) | 0.07517 |
> | (ii) | Equiformer + EMPP (300 epochs) | 0.03912 |
>
>
> **To evaluate the impact of EMPP on molecules with different sizes, we conducted experiments by categorizing the QM9 training data into three groups based on the number of atoms: (0-16), (17-19), and (20+).** These categories contain roughly equal amounts of data. In each experiment, we computed the EMPP loss using only molecules from one of these categories. As shown in the below table, the results demonstrate that EMPP consistently improves performance across different molecular sizes, with larger molecules experiencing more significant performance gains.
>
> | Index | Method      | HOMO | LUMO |
> | ----------- | ----------- | ----------- | ----------- |
> | (i) | Baseline (Using EMPP on all data)  | 0.041 | 14.2 |
> | (ii) | (0-16)  | 0.044 | 14.9 |
> | (iii) | (17-19)   | 0.043  | 14.4 |
> | (iv) | (20+)  | 0.043  | 14.5 |

---

> ### Comment · Reviewer_cKne · 2024-11-28
>
> Thank you for your detailed and thorough responses to the questions and suggestions in my review. I greatly appreciate the effort you have put into addressing the raised concerns and providing additional experimental results. Your explanations clarify key aspects of EMPP's design and its theoretical underpinnings, as well as its practical strengths. The new experiments on GEOM-Drug and the analysis of performance scaling with molecular size are especially helpful in demonstrating EMPP's applicability to more complex molecular systems.
> Thank you once again for your dedication and comprehensive responses. I look forward to seeing the final version of your paper and its contributions to the field.
> Best regards.

---

### Author Response · Authors · 2024-11-22
**To Reviewers**

Dear Reviewers,

Thank you for your time and the positive evaluation of our manuscript. We have submitted the revised version, where we believe we have effectively addressed all of your concerns. We have highlighted the modified sections in red. Specifically, in the updated manuscript, we have: (a) We revised some expressions that were prone to causing confusion. (b) We included experiments with the GEOM-Drug dataset and some ablation studies.

As the discussion session is drawing to a close, we would like to confirm whether you have any remaining concerns. We sincerely appreciate your diligent efforts. Thank you once again.

## To Reviewer cKne:
Thank you for your review and suggestions. We have addressed your questions in our rebuttal. Additionally, we have supplemented our experiments with the GEOM-Drug dataset and conducted an analysis of training sets with molecules of different scales. **In the absolute energy prediction on GEOM-Drug, EMPP can produce significant performance improvements.**

[Updated in 11.25, We trained the Drug dataset for 300 epochs to achieve a stable performance. The results indicate that EMPP can enhance energy prediction performance by 48% under non-equilibrium conditions. We have incorporated this into the revised version.]

## To Reviewer Koif:
Thank you for your review. We have taken your advice and have added an analysis of time within the revised manuscript. This part has also been included in our rebuttal.

## To Reviewer DiCi:
Thank you for your review and prompt response. If there are any further questions, please do not hesitate to let us know.

## To Reviewer RrYB:
Thank you for your review. We have addressed all of your concerns in our rebuttal. To put it succinctly, regarding the two issues you are most concerned about, we have provided explanations at the beginning of our rebuttal **(Response to main concerns)**.

Best regards, Authors

---

### Author Response · Authors · 2024-11-30
**Thanks to All Reviewers**

Dear Reviewers,

As the rebuttal period comes to a close, we would like to take this opportunity to express our sincere gratitude to all reviewers. Your insightful questions and constructive suggestions during the review phase have been invaluable. **We have carefully addressed all of your concerns in the rebuttal stage and have made corresponding updates to the paper.**

In the revised manuscript, we have made the following improvements:

1.We added experiments with GEOM-Drug, demonstrating significant improvements when EMPP (our method) is used as an auxiliary task.

2.We add ablation studies on EMPP, covering aspects such as time consumption, smooth distribution, molecular complexity and backbone models, which help to validate the design rationale behind EMPP.

3.We have modified some expressions to facilitate better understanding for the readers.

Your reviews have been instrumental in refining our work, and we are deeply grateful for the time and effort each of you has devoted to reviewing the paper and engaging in thoughtful discussions. **If you have any further questions, please do not hesitate to tell me, and we will reply quickly.**

Best regards, Authors

---

### Public Comment · ~Ameya_Daigavane1 · 2025-02-16
**Comparison to Symphony?**

Hi, nice paper! Thank you for citing our Symphony paper: https://openreview.net/forum?id=MIEnYtlGyv.
I would greatly appreciate a more detailed discussion of our work; since Symphony also define a masking strategy + use equivariant GNN embeddings + spherical harmonic projections to fill in atom positions conditional on neighborhood context. I understand the underlying task is different, but it seems unfair to not discuss these similarities.

Thank you,
Ameya

---

> ### Public Comment · ~Junyi_An1 · 2025-02-18
> **Comparisons Have Been Added**
>
> Hi Ameya,
>
> Thank you for your suggestions. We've included a comparison in the camera-ready version. A general comparison can be found at the end of Section 3.2.2 (POSITION PREDICTION) and in Section 4 (RELATED WORK). Due to space limitations, more detailed comparisons are provided in Appendix B.5. In this version, only the comparison with Symphony has been added, while the rest remains unchanged.
>
> Best regards,
>
> Authors

---

### Meta-Review · Area_Chair_AUQv · 2024-12-18

**Metareview:**

The submission presents a molecular representation self-supervised learning method. In contrast to existing ones, it asks the model to directly predict the position of a masked atom without given a noised position of it, in hope to avoid superficial mode width around the equilibrium structure in denoising-based methods without Boltzmann distribution samples. The paper makes a good summary on existing methods, and is technically sound. Experimental results support the motivation to improve upon denoising-based pretraining methods.

While the reviewers generally appreciate the mentioned contributions, they also raised concerns and insufficiencies, including limited evaluation cases, alignment in comparison settings (e.g., starting with pre-trained weights, architecture differences), discussion on computational overhead, and difference with denoising-based pretraining methods. In the rebuttal, the authors have provided more experimental results on GEOM-Drug for enriching evaluation, worked on more detailed alignment on comparisons with baselines, provided time consumption which seems reasonable and acceptable, and made further explanations on the conceptual relation with denoising together with presentation modifications. These newly provided results have effectively addressed most of the concerns, and the two negatively rating reviewers have raised their scores to positive. In light of this, I recommend accept for this submission.

**Additional Comments On Reviewer Discussion:**

Reviewers appreciated the novelty and soundness of directly predicting the position of a masked atom, and that the empirical results have justified the method promising. Reviewers also raised concerns and insufficiencies, including limited evaluation cases, alignment in comparison settings (e.g., starting with pre-trained weights, architecture differences), discussion on computational overhead, and difference with denoising-based pretraining methods. In the rebuttal, the authors have provided more experimental results on GEOM-Drug for enriching evaluation, worked on more detailed alignment on comparisons with baselines, provided time consumption which seems reasonable and acceptable, and made further explanations on the conceptual relation with denoising together with presentation modifications. These newly provided results have effectively addressed most of the concerns, and the two negatively rating reviewers have raised their scores to positive. In light of this, I recommend accept for this submission.

---

### Decision · Program_Chairs · 2025-01-22

Accept (Poster)